# GAR: GENERATIVE ADVERSARIAL REINFORCEMENT LEARNING FOR FORMAL THEOREM PROVING

**Ruida Wang[1], Jiarui Yao[1], Rui Pan[1], Shizhe Diao[2], Tong Zhang[1]**
[1]University of Illinois Urbana-Champaign, [2]NVIDIA
{ruidaw,jiarui14,ruip4}@illinois.edu
shizhe.diao@gmail.com
tongzhang@tongzhang-ml.org

## ABSTRACT

Solving math problems through verifiable languages such as Lean has significantly impacted both the mathematics and computer science communities. Current state-of-the-art models are often trained with expensive online Reinforcement Learning (RL) or expert iteration. However, these approaches rely on fixed problem sets, which causes inefficient training and limits the model to tackle complex problems. To overcome these limitations, we propose **GAR**: *Generative Adversarial Reinforcement learning*, a comprehensive RL training framework that jointly trains the problem composer and solver in an adversarial loop. **GAR** introduces an implicit curriculum learning mechanism, which aligns task difficulty with the prover's evolving capability. It thereby improves the training efficiency and enables stronger performance of proving advanced theorems. Experiments show that with **GAR** training, Goedel-Prover-V2-8B and DeepSeek-Prover-V2-7B achieve an average relative improvement in pass@32 of **4.20%** on MiniF2F-Test benchmark, while DeepSeek-Prover-V2's pass@32 on ProofNet-Test increases from 22.58% to **25.81%**. Beyond formal proving, **GAR** establishes a general RL paradigm for co-evolution of problem generation and solving under verifiable environments. The training code for this paper is open-sourced in https://github.com/RickySkywalker/GAR-Official

## 1 INTRODUCTION

The capability to perform formal mathematical reasoning has long been regarded as both a hallmark of human intelligence and a key objective of machine learning (Newell & Simon, 1956). The ability is typically assessed through rigorous mathematical derivations (Yang et al., 2024b). With the emergence of Large Language Models (LLMs), developing accurate and reliable reasoning has become an active area of research. Recent progress in ZERO RL training (Guo et al., 2025) has further advanced reasoning systems by introducing Long Chain-of-Thought (CoT) thinking models that have self-reflection and self-correction capability.

However, the inherent ambiguity of Natural Language (NL) makes it challenging to verify intermediate reasoning steps. This problem is more severe in advanced mathematics, where the task is to prove theorems rather than give a numerical or formulaic answer. The increasing complexity of modern math compounds this difficulty, as illustrated by the prolonged verification of Fermat's Last Theorem (Wang et al., 2024). To address this issue, researchers have grounded reasoning in formal logical systems, enabling automatic verification through Formal Language (FL). Based on this idea, some researchers model the reasoning process formally with dependent type languages like Lean (De Moura et al., 2015; Moura & Ullrich, 2021) and Coq (Coq, 1996). Other uses higher-order logic to build language like Isabelle (Paulson, 1994) and HOL (Harrison, 2009). All the above languages make every intermediate step of math reasoning verifiable.

Nevertheless, mastering FLs requires considerable expertise, particularly in dependent-type systems like Lean, where proofs often demand complex type matching and the use of functions with limited data (Wang et al., 2025c). Thus, many works have sought to leverage advances in LLMs to solve FL problems and train specialized FL provers (Polu et al., 2022; Jiang et al., 2022; Xin et al., 2024a;

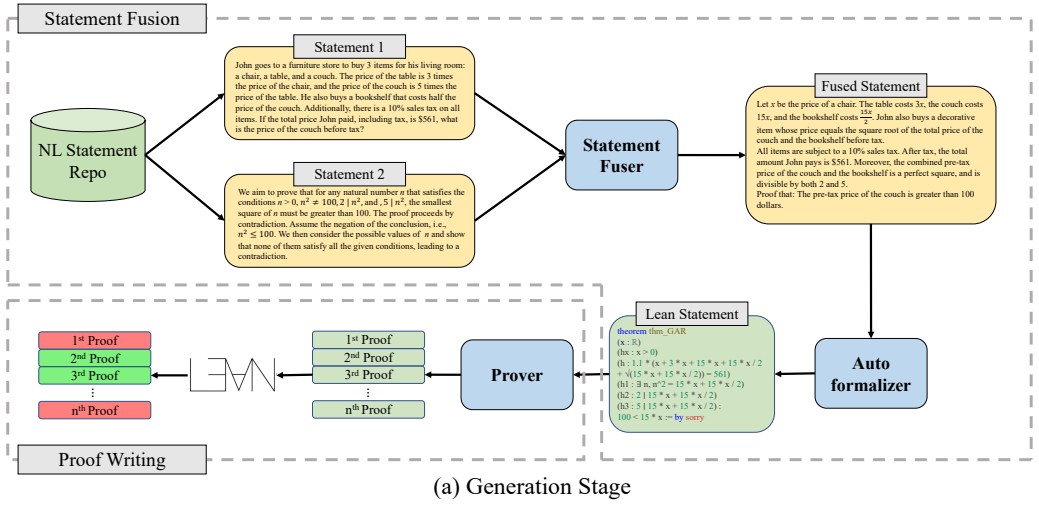

(a) Generation Stage

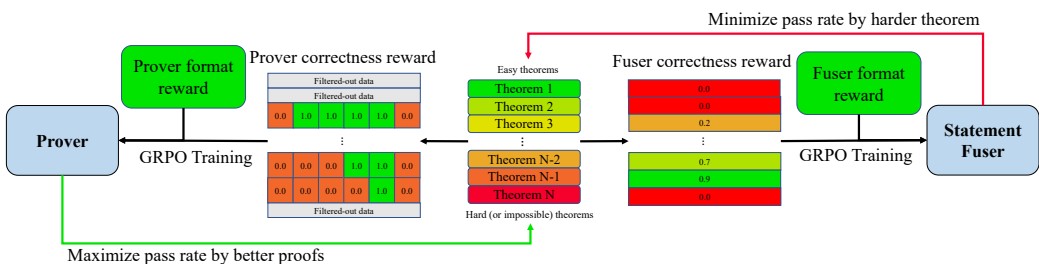

(b) Adversarial Reinforcement Learning Stage

Figure 1: **GAR Training Framework:** Each iteration of **GAR** consists two stages. (a) *Generation Stage*: Pairs of NL statements are sampled from the base repository and combined by the statement fuser to create more challenging problems that fit the current model's capability. Then, these statements are autoformalized and submitted to the prover to write multiple proofs. Subsequently, the proofs are checked by the Lean verifier for reward assignments. (b) *Adversarial Reinforcement Learning*: The prover is rewarded for producing correct proofs on medium and high-difficulty statements, while the statement fuser is rewarded for generating harder but solvable problems. This adversarial dynamic drives both models to evolve together.

Wang et al., 2024; Lin et al., 2025a; Dong & Ma, 2025b). The verifiability of FLs also motivates large-scale synthesis of new statements, which enables extensive expert iteration (Polu et al., 2022; Xin et al., 2025) or Reinforcement Learning (RL) (Ren et al., 2025; Wang et al., 2025a; Lin et al., 2025b) to further enhance perver's performance.

However, most prior approaches in RL or expert iteration rely on fixed theorem statement sets and optimize only the prover. The statement dataset is unchanged during the process. It also leads to wasted computation on trivial or unsolvable tasks. In expert iteration, datasets often require repeated annotation. On the other hand, advanced RL methods suffer from the absence of an adaptive difficulty level of statements during rollouts, which restricts concentrated exploration and limits progress on complex theorems. More broadly, only a few works discuss a systematic way to align problem difficulty with the prover's growing skill.

To address this limitation, we propose **GAR**: *Generative Adversarial Reinforcement Learning*, a comprehensive RL training framework as presented in Figure 1. **GAR** jointly optimizes the prover and the problem composer, termed as the statement fuser, through adversarial training. This process establishes an implicit curriculum learning that dynamically adjusts statement difficulty to match the prover's development. Each iteration of the **GAR** framework consists of two stages: the generation and the adversarial RL stage. In the generation stage, the statement fuser synthesizes harder statements from existing solvable ones, and the prover attempts to solve them. In the RL training stage, the fuser is rewarded for producing difficult yet solvable statements, while the prover is re-

warded for correctly proving medium and high-difficulty problems. This adversarial setup ensures that statement complexity evolves alongside the prover's capabilities, resulting in more efficient and effective training.

We summarize our contributions as follows: (1) We introduce **GAR**, a comprehensive RL training framework that establishes implicit curriculum learning, improving the prover's reasoning ability while preventing wasted effort on trivial or unsolvable statements. (2) We present *Statement Fusion* technique, which enables the creation of novel formal statements beyond direct formalization of NL problems, producing theorem statements better aligned with model capability. (3) We demonstrate the effectiveness of the **GAR** through extensive experiments by training two base models using **GAR**. We achieve an average of **4.20%** relative improvement on MiniF2F-Test (Zheng et al., 2021) and raise DeepSeek-Prover-8B's pass@32 on the more challenging ProofNet-Test (Azerbayev et al., 2023) from 22.58% to **25.81%**. It highlights the effectiveness and generality of **GAR**.

Furthermore, the **GAR** contributes a general RL paradigm for the co-evolution of problem generation and solving in verifiable environments, which offers new directions for adversarial co-training in other reasoning-intensive domains. To facilitate the advancement of the field, we will open-source the training and inference code of **GAR** in the near future.

## 2 METHODOLOGY

In this section, we introduce the **GAR** framework in detail. The prover's task is to produce Lean4 proofs from NL-FL statement pairs. Our framework trains the prover to improve at the task by adapting theorem difficulty to the prover's evolving capabilities through adversarial training of both the prover and the statement fuser. Each iteration of **GAR** consists of two stages: the *Generation stage* (Section 2.1) and the *Adversarial Reinforcement Learning stage* (Section 2.2). We first describe the behavior of each stage in a single iteration in detail, then present the complete procedure in Section 2.3.

### 2.1 GENERATION STAGE

The generation stage of **GAR** consists of two processes, namely statement fusion, which produces new statements, and proof writing, which generates and evaluates candidate proofs.

#### 2.1.1 STATEMENT FUSION

This process generates harder statements from existing ones. It aims to mitigate the mismatch between the fixed datasets and the prover's evolving ability. This process first samples two NL statements from the base dataset, formally: $\boldsymbol{s}_{base} = (s_1^{(NL)}, s_2^{(NL)}) \sim \mathcal{D}_{stat}$, where $\mathcal{D}_{stat}$ is a repository consists of 793,243 NL statements collect from Lean-Workbook (Wu et al., 2024b) and Numina-Math (LI et al., 2024). The sampled pair is then passed to the statement fuser trained in the previous iteration of **GAR**. The fuser is instructed to synthesize a more challenging statement by combining the key ideas of the two inputs, namely:

$$s^{(NL)} = Fuser_{i-1}(\boldsymbol{s}_{base})$$

where $i$ denotes the current iteration index, $s^{(NL)}$ is the generated NL statement, and $Fuser_{i-1}$ is the fuser from the prior step (initialized as base model when $i = 0$).

In particular, we chose thinking models like Qwen3 (Team, 2025) or DeepSeek-R1 (Guo et al., 2025) as the base model for the statement fuser because of its outstanding performance. However, direct use of their native Long CoT capability often results in overthinking and degraded quality of the generated statement. To mitigate this, we reinitialize the thinking process by passing the default thinking stage and restarting it with a dedicated indicator token named <analysis>. This trick yields more focused and task-specific reasoning. The full prompt for the statement fusion is shown in Figure 2. Subsequently, the generated NL statement is passed to the autoformalizer, which converts the NL statement into Lean statement $s^{(FL)}$. The formalized statement is then passed to Lean4 for compilation check. This step filters out statements with grammatical errors.

In the statement fusion stage, they deliberately separate the fusion of NL statements from the formalization of FL statements. Two key insights guide this design. First, prior studies (Wang et al., 2024;

> You are an expert mathematics educator skilled in problem design. Your task is to combine multiple given problems into a SINGLE, more challenging problem by combining their key elements. Follow these steps: Please first do the following steps as your analysis process enclosed within <analysis></analysis>.
> 1. Analyze the points of knowledge that need to be used when solving the proof problem and identify overlapping or complementary aspects (e.g., shared topic areas or contrasting difficulty levels).
> 2. Draft the new problem that integrates at least 2 key components from each original problem and make sure the new problem requires multi-step reasoning (e.g., combining algebraic manipulation with probabilistic analysis). Also, your combined problem should have no non-trivial extension.
> 3. Additionally, you should make sure that the new problem is solvable.
> After your analysis, you should put the new problem into an MD code block. The new problem should be a SINGLE proof problem.
> Here are the statements you need to fuse:
> Problem 1: <statement_1>
> Problem 2: <statement_2>

Figure 2: Prompt for the Statement Fuser to generate harder statements from existing ones and restart thinking using a new indicator in **GAR** generation stage

Xin et al., 2024b) have shown that NL statements play a crucial role in supporting formal reasoning. Secondly, current 8B-scale general LLMs exhibit only a limited understanding of FL. This causes them to fail to capture essential relations, which hinders the generation of more robust statements. Consequently, the fuser produces statements that have a high likelihood of failing the compilation check if we directly fuse formal statements, as seen in Dong & Ma (2025b). Thus, **GAR** firstly fuses the statement in NL and formalizes it to FL, ensuring the fused statements are adaptively more challenging than the base problems to fit prover performance evolution.

### 2.1.2 PROOF WRITING

Utilizing the NL-FL statement pair generated by the fusion process, the prover from the last iteration generates $n$ candidate proofs (with $n = 16$ in our implementation), denoted as $\{o_k\}_{k=1}^n$, specifically:

$$o_k = Prover_{i-1}(s^{(NL)}, s^{(FL)}), \forall k \in [1, n]$$

When $i = 0$, it is the base prover model like deepSeek-Prover-V2 (Ren et al., 2025) or Goedel-Prover-V2 (Lin et al., 2025b).

The set of candidate proofs $\{o_k\}_{k=1}^n$ is then passed to the Lean4 verifier to check correctness. Based on the verification results, we compute the empirical pass rate $p$ of this statement for reward assignment as $p = \frac{n_{pass}}{n}$, where $n_{pass}$ is the number of proofs that pass verification.

## 2.2 ADVERSARIAL REINFORCEMENT LEARNING

This section outlines the training process of **GAR**, where the prover and statement fuser are optimized in competition, deriving adversarial learning and mutual improvement.

### 2.2.1 STATEMENT FUSER TRAINING

The statement fuser is trained to synthesize problems that are slightly beyond the prover's current capability, thereby creating an implicit curriculum. To achieve this, we assign rewards to the generated NL statements that encourage the production of more challenging but solvable problems. We adopt a variant of Group Relative Policy Optimization (GRPO) (Shao et al., 2024) for this optimization. Specifically, for each step $i$, the fuser generates $N$ statements (with $N = 1024$ in our implementation). Formally, generated statement set is: $\left\{ s_{i,j}^{(NL)} \right\}_{j=1}^N$, and base statement set is represented as: $\left\{ \boldsymbol{s}_{i,j}^{(base)} = \left( s_{i,j,1}^{(NL)}, s_{i,j,2}^{(NL)} \right) \right\}_{j=1}^N$. The statement fuser $Fuser_{i-1}$ is then updated by maximizing the following objective.

$$\mathcal{J}_{GRPO}^{(F)}(\theta) = \mathbb{E}_{\left[\left\{\boldsymbol{s}_{i,j}^{(base)}\right\}_{j=1}^N \sim \mathcal{D}_{stat}; \left\{s_{i,j}^{(NL)} \sim \pi_{\theta_{old}}\left(\boldsymbol{s}_{i,j}^{(base)}\right)\right\}\right]}$$

$$\frac{1}{N} \sum_{j=1}^N \left(\min\left\{\delta_{i,j}^{(stat)} A_{i,j}^{(stat)}, \text{clip}\left\{\delta_{i,j}^{(stat)}, 1 \pm \varepsilon\right\} A_{i,j}^{(stat)}\right\} - \beta \mathbb{D}_{KL}(\pi_\theta \| \pi_{\theta_{ref}})\right), \tag{1}$$

$$\delta_{i,j}^{(stat)} = \frac{\pi_\theta\left(s_{i,j}^{(NL)} | \boldsymbol{s}_{i,j}^{(base)}\right)}{\pi_{\theta_{old}}\left(s_{i,j}^{(NL)} | \boldsymbol{s}_{i,j}^{(base)}\right)}, \tag{2}$$

$$\mathbb{D}_{KL}(\pi_\theta \| \pi_{\theta_{ref}}) = \frac{\pi_{\theta_{ref}}\left(s_{i,j}^{(NL)} | \boldsymbol{s}_{i,j}^{(base)}\right)}{\pi_\theta\left(s_{i,j}^{(NL)} | \boldsymbol{s}_{i,j}^{(base)}\right)} - \log \frac{\pi_{\theta_{ref}}\left(s_{i,j}^{(NL)} | \boldsymbol{s}_{i,j}^{(base)}\right)}{\pi_\theta\left(s_{i,j}^{(NL)} | \boldsymbol{s}_{i,j}^{(base)}\right)} - 1 \tag{3}$$

where $\pi_\theta$ denotes the policy model with parameter $\theta$. It represents the statement fuser here. $\theta_{ref}$ is the parameter for base statement fuser, $\theta_{old}$ is the parameter for the fuser in the previous step, $\varepsilon$ and $\beta$ are hyper-parameters, and $A_{i,j}$ is the advantage, computed from the reward set $\boldsymbol{r}_i^{(stat)} = \{r_{i,j}^{(stat)}\}_{j=1}^{N_i}$ by:

$$A_{i,j}^{(stat)} = \frac{r_{i,j}^{(stat)} - \text{mean}(\boldsymbol{r}_i^{(stat)})}{\text{std}(\boldsymbol{r}_i^{(stat)})}, \quad r_{i,j}^{(stat)} = (1 - p_{i,j}) \cdot (1 - m_{i,j}) \cdot \mathbb{I}\{p_{i,j} \neq 0\} \tag{4}$$

where $p_{i,j}$ is the prover's empirical pass rate on $s_{i,j}^{(FL)}$, and $m_{i,j}$ is the statement modification rate, which indicates the portion of proofs that the prover tries to modify its statement. The reward is set to 0 if the prover fails to solve the problem, which indicates the statement is too difficult or unsolvable.

We introduce the term $1 - m_{i,j}$ as a soft statement modification penalty to balance the risk between reward hacking and the need to preserve model capability. Because of Long CoT training, current expert provers often acquire strong self-correction capability. While valuable, this ability can lead the model to change formal statements during proof writing. This may lead to serious reward hacking if unconstrained. Conversely, a strict ban on such modifications would suppress self-correction and reduce proofreading accuracy. Our soft penalty discourages excessive statement change without too much harm to the result.

In summary, the training design for the statement fuser rewards the model to lower the prover's pass rate by composing more challenging statements.

### 2.2.2 PROVER TRAINING

To achieve adverasrial training, the prover is optimized to maxmize the pass rate on generated statements. We employ a variant of the GRPO algorithm for such training. Specifically, for each theorem statement $\boldsymbol{s}_{i,j} = (s_{i,j}^{(NL)}, s_{i,j}^{(FL)})$, we update the prover model $Prover_{i-1}$ by maximizing the following objective function:

$$\mathcal{J}_{GRPO}^{(P)}(\omega) = \mathbb{E}_{\left[\{o_{i,j,k}\}_{k=1}^n \sim \pi_{\omega_{old}}(\boldsymbol{s}_{i,j})\right]}$$

$$\frac{1}{n} \sum_{k=1}^n \left(\min\left\{\delta_{i,j,k}^{(pr)} A_{i,j,k}^{(pr)}, \text{clip}\left\{\delta_{i,j,k}^{(pr)}, 1 \pm \varepsilon\right\} A_{i,j,k}^{(pr)}\right\} - \beta \mathbb{D}_{KL}(\pi_\omega \| \pi_{\omega_{ref}})\right), \tag{5}$$

$$\delta_{i,j,k}^{(proof)} = \frac{\pi_\omega\left(o_{i,j,k} | \boldsymbol{s}_{i,j}\right)}{\pi_{\omega_{ref}}\left(o_{i,j,k} | \boldsymbol{s}_{i,j}\right)} \tag{6}$$

where $\pi_\omega$ is the prover as policy model, $\omega$ is the parameter of prover, and $A_{i,j,k}$ is the advantage of proof $o_{i,j,k}$, computed from the reward set $\boldsymbol{r}_{i,j}^{(proof)} = \{r_{i,j,k}^{(proof)}\}_{k=1}^n$ by:

$$A_{i,j,k}^{(proof)} = \frac{r_{i,j,k}^{(proof)} - \text{mean}(\boldsymbol{r}_{i,j}^{(proof)})}{\text{std}(\boldsymbol{r}_{i,j}^{(proof)})}, \quad r_{i,j,k} = 1 - 0.5 \cdot m_{i,j,k} \tag{7}$$

where $m_{i,j,k}$ is a binary indicator of whether a statement modification occurred in proof $o_{i,j,k}$. Similar to the training of the statement fuser, modifications are penalized but not strictly prohibited. Furthermore, to ensure the prover is trained on high-quality data, we exclude statements with an empirical pass rate of $0$ (unsolvable) or above $0.5$ (too easy) following experience in Wang et al. (2025a); Dong & Ma (2025b). It make sure the prover is only trained with hard and medium-level problems. This optimization scheme drives the prover to compete with the statement fuser by continually enhancing its proof-generation capability.

## 2.3 SUMMARY

The **GAR** is an iterative framework (Wang et al., 2023) that loops the generation stage and the RL stage details above. To provide a clear overview, we present the complete framework in the form of pseudo-code as follows:

---
**Algorithm 1 GAR**

---
**Require:** $\mathcal{D}_{stat}$, Statement Fuser (base) as $Fuser$, Prover (base) as $Prover$, Autoformalizer as $AF$

1: $\pi_\omega, \pi_{\omega_{old}}, \pi_{\omega_{ref}} = Prover$; $\pi_\theta, \pi_{\theta_{old}}, \pi_{\theta_{ref}} = Fuser$
2: **for** $i : [1, T]$ **do**                                                          ▷ Step $i$ of **GAR**
3:     $\{\boldsymbol{s}_{i,j}^{(base)}\}_{j=1}^N \sim \mathcal{D}_{stat}$                  ▷ Sample base statements
4:     $\{s_{i,j}^{(NL)} \sim \pi_\theta(\boldsymbol{s}_{i,j}^{(base)})\}_{j=1}^N$          ▷ Fuse NL statements
5:     $\{s_{i,j}^{(FL)} \sim AF(s_{i,j}^{(NL)})\}_{j=1}^N$                                ▷ Autoformalize statement
6:     $\{\boldsymbol{s}_{i,j} = (s_{i,j}^{(NL)}, s_{i,j}^{(FL)})\}_{j=1}^N$
7:     **for** $j : [1, N]$ **do**
8:         $\{o_{i,j,k} \sim \pi_\omega(\boldsymbol{s}_{i,j})\}$                          ▷ Generate proofs
9:         Obtain $p_{i,j}, m_{i,j}$ by Lean checking of proofs
10:    $\pi_\theta = \pi_{\theta_{old}} \leftarrow Optimize(\mathcal{J}_{GRPO}^{(F)}(\theta))$   ▷ Train NL Fuser
11:    $\pi_\omega = \pi_{\omega_{old}} \leftarrow Optimize(\mathcal{J}_{GRPO}^{(P)}(\omega))$   ▷ Train Prover
12: **return** $\pi_\theta, \pi_\omega$

---

When trained in multiple iterations, the **GAR** establishes an implicit curriculum by aligning statement difficulty with the prover's evolving capability. The statement fuser is trained to reduce the pass rate by generating more challenging statements, while the prover is optimized to increase it by producing more valid proofs. Together, their adversarial interaction drives progressive improvement.

## 3 EXPERIMENTS

We conduct comprehensive experiments on the MiniF2F-Test (Zheng et al., 2021) and ProofNet-Test (Azerbayev et al., 2023) benchmarks to assess the performance of the **GAR** framework in formal proof writing. Specifically, we show in Section 3.3 that the models trained with **GAR** achieve better empirical results, confirm in Section 3.4 that adversarial training induces an implicit curriculum by generating progressively harder statements, and report ablation study results in Section 3.5. Due to space limitations, we have included the efficiency study and case study in Appendix D.

## 3.1 EXPERIMENT SETUP

### 3.1.1 DATASET AND TASK

We measure the Lean4 reasoning capability of the **GAR** trained model by MiniF2F-Test (Zheng et al., 2021) and ProfNet-Test (Azerbayev et al., 2023) benchmarks. They are challenging benchmarks and adopted in nearly all major studies in the field (Xin et al., 2024b; Lin et al., 2024; Wang et al., 2024; Wu et al., 2024a; Polu et al., 2022; Dong & Ma, 2025a; Lin et al., 2025b; Wang et al., 2025c).

The MiniF2F-Test benchmarks comprise 244 Lean4 statements, spanning from high school competition problems to elementary undergraduate-level theorem proofs. Specifically, MiniF2F-Test

Table 1: Main experimental results of **GAR** trained models compared to recent provers models.

| Method | Sample budget | MiniF2F-Test | ProofNet-Test |
|---|---|---|---|
| Lean-STaR (Lin et al., 2024) | $64 \times 1 \times 50$ | 46.31% | - |
| InternLM-2.5-StepProver (Wu et al., 2024a) | $4 \times 32 \times 600$ | 50.70% | 18.80% |
| DeepSeek-Prover-V1.5-RL (Xin et al., 2024b) | 128 | 50.00% | 18.20% |
| STP-Lean (Dong & Ma, 2025b) | 128 | 56.15% | 19.50% |
| MA-LoT (Wang et al., 2025c) | 32 | 61.07% | 15.47% |
| Kimina-Prover-Distill-7B (Wang et al., 2025a) | 32 | 63.10% | - |
| DeepSeek-Prover-V2-7B (Ren et al., 2025) | 32 | 70.49% | 22.58% |
| Geodel-Prover-V2-8B (Lin et al., 2025a) | 32 | 77.87% | - |
| *Our models* | | | |
| GAR on Deepseek-Prover-V2 | 32 | **74.18%** | **25.81%** |
| GAR on Goedel-Prover-V2 | 32 | **80.33%** | - |

Table 2: Average proof correctness rate for Goedel-Prover-V2-8B (base model) and GAR model trained on base model.

| Step idx | Base Model | GAR model |
|---|---|---|
| 0 | 29.16% | 29.16% |
| 1 | 16.50% | 23.71% |
| 2 | 11.58% | 20.53% |
| 3 | 7.61% | 20.08% |
| 4 | 7.69% | 21.79% |

Table 3: Statement modification rate for dropping statement modification penalty and full GAR trained models.

| Step idx | w/o Stat. Penalty | Full GAR |
|---|---|---|
| 0 | 42.94% | 42.94% |
| 1 | 48.18% | 48.72% |
| 2 | 60.42% | 30.50% |
| 3 | 71.82% | 39.65% |
| 4 | 74.11% | 33.63% |

comprises problems formalized from the MATH dataset (Hendrycks et al., 2021), high school competitions such as AMC, AIME, and IMO, as well as self-crafted problems. ProofNet-Test consists of 186 theorems formalized from standard undergraduate textbooks on advanced topics such as real and complex analysis, linear algebra, abstract algebra, and topology. In our setting, we train the LLM with **GAR** to generate Lean4 proofs from the NL-FL statement pair. To avoid overloading the model, all the imports and namespaces are manually configured.

### 3.1.2 BASELINES

To evaluate the effectiveness of **GAR**, we compare it against strong open-source baselines, including Lean-STaR (Lin et al., 2024), InternLM-2.5-StepProver (Wu et al., 2024a), Kimina-Prover-Preview-Distill-7B (Wang et al., 2025a), DeepSeek-Prover-V1.5-RL (Xin et al., 2024b), STP-Lean (Dong & Ma, 2025a), MA-LoT (Wang et al., 2025c), Goedel-Prover-V2 (Lin et al., 2025b), and DeepSeek-Prover-V2 (Ren et al., 2025). For baseline models that require Long CoT reasoning (Kimina-Prover, MA-LoT, DeepSeek-Prover-V2, and Goedel-Prover-V2), we restrict the reasoning length to 16,384 tokens to conserve computational resources.[1]

### 3.2 IMPLEMENTATION DETAILS

In the generation stage, we construct a repository of 793,243 NL statements from Numina-Math (LI et al., 2024) and Lean-Workbook (Ying et al., 2024) datasets. For statement fusion, we employ Qwen3-8B (Team, 2025) as the base model for the statement fuser because of its skip-thinking capability. We apply Kimina-Autoformalizer-7B (Wang et al., 2025a) as the autoformalizer. For proof generation, we use DeepSeek-Prover-V2-7B (Ren et al., 2025) and Goedel-Prover-V2 (Lin et al., 2025b) as base models for provers. We sample 1,024 theorems per step and generate 16 proofs per theorem following (Wang et al., 2025a). We also restrict the sequence length to 16,384 tokens for both models. The GRPO hyperparameters are set with a learning rate of 2E-6, $\varepsilon = 0.2$, and $\beta = 0.01$. We perform three iterations of **GAR** training on Goedel-Prover-V2 and five iterations on DeepSeek-Prover-V2, which costs around 140 H100 hours for each training. On average, DeepSeek-Prover-V2 costs fewer hours per iteration due to average shorter thinking. In verification, any proof

---

[1]Because of this restriction, our reported results for DeepSeek-Prover-V2 and Goedel-Prover-V2 differ from those in the original papers, where the evaluations used a context length of 40,960.

involving statement modification or relying on the "sorry" or "admit" tactic is counted as incorrect to ensure fairness.

## 3.3 RESULTS

Table 1 demonstrates the empirical results of applying **GAR** to train base prover models. For Goedel-Prover-V2, MiniF2F-Test pass@32 raises to 80.33%, indicating a **3.16%** relative gain. For DeepSeek-Prover-V2, the MiniF2F-Test score improves to 74.18%, corresponding to a **5.23%** increase. On the more challenging ProofNet-Test benchmark, which targets advanced mathematics topics, **GAR** enhances DeepSeek-Prover-V2's pass@32 rate from 22.58% to **25.81%**.[2]

Compared with other baselines, both the base model and the **GAR**-trained models achieve consistent and substantial gains, highlighting that **GAR** can contribute to models that already have outstanding performance. Compared to base models, the improvements suggest that the adversarial method proposed in **GAR** enables provers to tackle increasingly difficult problems by establishing an implicit curriculum. Such a curriculum can progressively adapt problem difficulty to the model's capability, allowing the prover to explore deeper reasoning strategies and solve more advanced theorems. This behavior becomes more significant as the theorems get harder.

## 3.4 STATEMENT FUSER STUDY

To show that **GAR** produces increasingly more difficult statements across iterations and thereby establishes an implicit curriculum learning, we examine the difficulty of problems generated by the statement fuser. In each iteration, we randomly sample 50 generated statements and compute their average proof correctness rate using both the base prover and **GAR**-trained prover at that iteration. Details of the metric we used are provided in Appendix B.2. The result of this experiment is reported in Table 2.

From the results, we can observe that the base model exhibits a consistent performance degradation, with accuracy falling from 29.16% at the first iteration's data to 7.69% by the fifth. In contrast, the **GAR**-trained model shows only a minor initial decline due to the statement matching but stabilizes at around 21% across later iterations. These findings confirm that **GAR** progressively generates harder statements, as evidenced by the base model's decline. The maintained trained model's performance demonstrates the adversarial training strength of the prover's performance over time.

## 3.5 ABLATION STUDIES

### 3.5.1 EFFECT OF STATEMENT MODIFICATION PENALTY

We evaluate the impact of the statement modification penalty by monitoring statement modification rates across by the provers during training rounds. Details of this metric are provided in Appendix B.3. We trained a variant of **GAR** on Goedel-Prover-V2-8B without the penalty in both the statement fuser and prover, and compared it with the full **GAR**.

Table 3 shows that without the modification penalty, the prover exploits its self-correction ability by simplifying statements. Such behavior worsens as the training progresses; by the fourth step, 74% of statements were modified at least once, which is a clear signal of reward hacking. In contrast, with the penalty enabled, the modification rate remains below 40% throughout. This study confirms the effectiveness of our statement penalty in preventing reward hacking.

Table 4: Pass@32 results on MiniF2F-Test for Goedel-Prover-V2-8B and its GRPO and GAR trained version.

| Method | MiniF2F-Test |
|---|---|
| **Base model** | 77.87% |
| **GRPO trained** | 77.46% |
| **GAR trained** | **80.33%** |

### 3.5.2 COMPARE TO DIRECT RL TRAINING

This experiment tests whether jointly evolving the prover and statement fuser is more effective than training the prover alone using existing data. We conducted three additional GRPO iterations on

---

[2]We do not report Goedel-Prover-V2's accuracy on ProofNet due to the absence of the reference results in Lin et al. (2025b).

Goedel-Prover-V2-8B using the same sampling budget and formalized NL base dataset as **GAR**. The results are shown in Table 4. The **GAR**-tained model outperforms the variant trained with traditional GRPO. We conclude that such an improvement to **GAR** progressively raises problem difficulty, enabling the prover to handle more complex statements. In contrast, further RL training on static dataset degrades performance, as the base model is already heavily RL trained. These findings further support the generality of **GAR**, demonstrating is ability to enhance model's performance while standard RL no longer provides benefits.

## 4 RELATED WORK

### 4.1 LLM FOR FORMAL THEOREM PROVING

Recently, applying LLMs to support formal theorem proving has become a prominent research direction. Training approaches can be broadly divided into two categories: model trained solely with Supervised Fine-Tuning (SFT) and those combining SFT with Reinforcement Learning (RL). Early SFT-based provers include Expert Iteration (Polu & Sutskever, 2020), Re-Prover (Yang et al., 2024c), TheoremLlama (Wang et al., 2024), DeepSeek-Prover-V1 (Xin et al., 2024a), InternLM-2.5-StepProver (Wu et al., 2024a), MA-LoT (Wang et al., 2025c), and Goedel-Prover-V1 (Lin et al., 2025a). These models typically require multiple rounds of large-scale annotation with existing formal solvers, which demands a significant amount of computational resources and limits exploration. To further advance the prover, researchers began incorporating advanced RL techniques. For instance, DeepSeek-Prover-V1.5 (Xin et al., 2024b) employs DPO. After the "ZERO" RL technique that enables Long CoT thinking developed by Guo et al. (2025), provers like Kimina-Prover (Wang et al., 2025a), DeepSeek-Prover-V2 (Ren et al., 2025), and Goedel-Prover-V2 (Lin et al., 2025b) enable models to produce better formal proof after thinking. However, existing RL approaches still rely on a fixed collection of statements, preventing statements from adapting to the prover's evolving skill. In contrast, **GAR** jointly trains a statement fuser and a prover, ensuring that generated theorems remain suitably challenging as the model improves.

### 4.2 RL METHODS FOR LLM

RL has been central to the development of reasoning LLMs (Xiong et al., 2025). Early efforts such as DeepSeek-Math (Shao et al., 2024) and Qwen-2.5-Math (Yang et al., 2024a) applied reward modeling with GRPO algorithm to enhance exploration. The release of OpenAI-O1 (OpenAI, 2024) and DeepSeek-R1 (Guo et al., 2025) demonstrated that verifier-based rewards in RL can enable complex reasoning strategies, including backward search and self-correction. Such techniques have been widely adopted in systems like Qwen3 (Team, 2025) and Kimi-K2 (Team et al., 2025). Nevertheless, nearly all prior work uses the verification signal only to optimize the problem solver, leaving the problem composer untrained. On the other hand, **GAR** simultaneously improves both statement fuser and the prover, creating an implicit curriculum that adapts task difficulty and allows the model to acquire more complex reasoning skills step-by-step.

## 5 CONCLUSION

This paper presents **GAR**, a comprehensive training framework for formal theorem proving. **GAR** aims to resolve the inefficient and suboptimal performance caused by traditional expert iteration and online RL in prover training. **GAR** achieves more efficient training by using statement fusion to formulate statements in the RL process to avoid the prover from annotating problems beyond its capability range. After **GAR** training, provers can solve more advanced problems through the implicit curriculum learning that enables the model to explore further. Furthermore, **GAR** can improve the performance of models that have been through heavy RL training by letting the model explore more difficult statements during the training process. Experiments of applying **GAR** to DeepSeek-Prover-V2 and Goedel-Prover-V2 yield an average relative improvement of **4.20%** on the MiniF2F-Test dataset and improve the DeepSeek-Prover-V2's performance on ProofNet-Test from 22.58% to **25.81%**. Beyond theorem proving, **GAR** offers a general RL paradigm of co-evolution of the problem generation and solving under a verifiable environment. It provides a foundation for adversarial co-training in other reasoning-intensive domains.

ETHICS STATEMENT

After carefully reviewing the ethical regulations of the conference, to the best of our knowledge, this work does not present any foreseeable ethical concerns. This research focuses exclusively on formal theorem proving using publicly available mathematical datasets, without involving human subjects, private data, or sensitive content. As far as we are concerned, no negative societal or ethical impacts are anticipated for the contribution of this work. We only use LLMs to polish the writing style and fix grammatical errors in the paper.

REPRODUCIBILITY STATEMENT

We have made efforts to ensure that our work is reproducible. The detailed description of the **GAR** framework, including pseudocode, data source, reward definitions, and training objectives, is presented in Section 2. The experimental setup, including benchmarks, baselines, base model choice, and hyperparameters, is presented in Section 3.1 and Section 3.2. We also plan to further ensure the reproducibility by open-sourcing the code in the near future.

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

## A  ACKNOWLEDGMENT

This material is based upon work supported partially by NSF under Grant No. 2416897, Grant No. 2505932, and by ORN under Grant No. N000142512318. This research used both Delta (NSF award OAC 2005572) and DeltaAI (NSF award OAC 2320345) advanced computing systems, and computing resources provided by Illinois Computes and NAIRR Pilot NAIRR250157.

## B  DETAILS OF THE METRICS

For completeness, this section provides the metric definitions that were omitted due to space limitations in Section 3.

### B.1  PASS RATE

Pass@$x$ is a widely used metric for evaluating formal theorem provers (Polu et al., 2022; Jiang et al., 2022; Wang et al., 2024; Xin et al., 2024a; Dong & Ma, 2025b; Wang et al., 2025b; Lin et al., 2025b; Wu et al., 2024a; Wang et al., 2025a; Li et al., 2025; Yao et al., 2025; Zhang et al., 2026). For each theorem statement, the model generates $x$ candidate proofs, which are further checked by the Lean4 verifier. If at least one of the $x$ candidates is correct, the theorem is counted as a proved theorem. The pass@$x$ of the LLM is the fraction of theorems in the dataset with at least one correct proof generated by the prover.

### B.2  AVERAGE PROOF CORRECTNESS RATE

This metric measures the relative difficulty of a set of statements for a given prover, which is used in Section 3.4). Let $\mathcal{D}$ be a dataset of size $m$. For each theorem $t_i \in \mathcal{D}$, we sample 16 proofs from the prover and record the number of proofs that pass the verification, which is $p_i$. The average proof correctness rate is calculated by

$$\frac{\sum_{i=1}^{m} p_i}{16 \cdot m}$$

A high value indicates the dataset is easier for the given prover.

Note that in this experiment, to demonstrate the significance of **GAR** better, we train two extra rounds of Goedel-Prover-V2-8B model.

### B.3  STATEMENT MODIFICATION RATE

Since current advanced provers have Long Chain-of-thought (CoT) capability, which enables self-reflection, backtracking, and self-correction. In the Long CoT process, a prover may alter the original statement into a simplified variant in its reasoning trace and ultimately produce a proof for the modified version. To quantify this behavior, we define statement modification rate as the portion of theorems in a dataset that have at least one statement modification across 16 generated proofs. This metric helps assess how often the prover attempts to modify the problem and whether such a behavior may be too significant to cause severe reward hacking.

## C  DIRECT COMPARISON WITH RELATED WORKS

This section aims to provide a more direct comparison between **GAR** and other works that also try to build a dynamic dataset during training, namely Goedel-Prover-V2 (Lin et al., 2025b) and STP (Dong & Ma, 2025b).

**Compared to Goedel-Prover-V2:**  The statement generation in Goedel-V2 relies on a frozen large model to synthesize data for SFT. Crucially, during the RL phase, their statement set remains static. This lack of dynamic updates of statements based on the prover's evolving capabilities may lead to suboptimal performance and efficiency as the prover outpaces the fixed problem set. In contrast, GAR continuously updates the statement fuser via adversarial RL, ensuring the generated statements

Table 5: pass@32 results on PutnamBench

| Model Type | PutnamBench |
|---|---|
| **DeepSeek-Prover-V2-7B (Ren et al., 2025)** | 22/660 |
| **GAR DeepSeek-Prover** | 24/660 |

Table 6: Statement Modification Rate for MiniF2F-Test under pass@32

| Model Type | Base Model | GAR trained |
|---|---|---|
| **DeepSeek-Prover-V2 (Ren et al., 2025)** | 6.96% | 13.11% |
| **Goedel-Prover-V2 (Lin et al., 2025b)** | 24.18% | 27.05% |

remain progressively challenging and aligned with the prover's current skill level during the RL phase, leading to better empirical performance.

**Compared to STP:** While STP trains a conjecture model based on the prover's feedback, it relies on the expert iteration framework based on offline SFT. This process is inherently inefficient because it requires generating a massive volume of training data in a single iteration to achieve effective SFT training. According to Dong & Ma (2025b), they generate 75,000 conjectures per iteration, which is larger than the entire statement set for our generation. Conversely, GAR operates within an online RL cycle. This allows the fuser and prover to continuously update more efficiently, achieving superior performance without the computational burden of the large-scale per-iteration data required by STP.

# D ADDITIONAL EXPERIMENTS

This appendix section provides additional experiments that are omitted in the main paper due to space limitations.

## D.1 RESULTS ON PUTNAMBENCH

Given that Lean4 formal reasoning is a rapidly progressing field, the MiniF2F-Test serves as a benchmark with high accuracy in current models. To further validate that the **GAR** training can also make the model perform better on more advanced benchmarks, we evaluate both DeepSeek-Prover-V2-7B and GAR-trained DeepSeek-Prover on PutnamBench under pass@32. The results are demonstrated in Table 5.

We can see that the GAR-trained model solves four additional problems compared to the base model on this challenging benchmark. This consistent improvement across MiniF2F, PutnamBench, and ProofNet demonstrates the robustness of the implicit curriculum established by **GAR**.

## D.2 EFFICIENCY STUDY

We analyze the computational efficiency of the **GAR** framework, as training from scratch demands substantial resources, to evaluate this, we compare our approach with Kimina-Prover (Wang et al., 2025a), selected for its transparent reporting of training details. The number of roll-out theorems and proofs per iteration in Wang et al. (2025a) matches our configuration. From Figure 4 in their work, we observe that Kimina-Prover achieves approximately 2% of performance gains after 25 training iterations, where **GAR** trained Goedel-Prover-V2-8B reaches more improvements in merely three iterations.

Notably, despite Kimina-Prover starting from an SFT base model, while **GAR** starts from a heavily RL-trained base model, this comparison strengthens our efficiency claim. Because it is well established in prior work (Guo et al., 2025) that models already heavily optimized via RL face diminishing returns and are inherently harder to improve than SFT baselines. The fact that **GAR** achieves relatively significant gains on top of RL-optimized base models in only 3-5 iterations, while Kimina-Prover gains approximately 2% from an SFT starting point over 25 iterations, demonstrates the high sample efficiency of our adversarial training paradigm. Given resource constraints that

prevent exhaustive experimentation, we leave the discussion of the scalability of **GAR** for future work.

### D.3 STATEMENT MODIFICATION RATE STUDY

To dive deeper into the behavior of statement modification, we provide detailed statement modification rates of both the base model and the **GAR** trained model in MiniF2F-Test under pass@32. The results are shown in Table 6

From the comparison of base models, we can see that the Goedel's increased statement modification rate happens together with its performance enhancement. We attribute this to the stronger self-correction capability it obtains. Similarly, the soft penalty in GAR ensures that the model is penalized if it simplifies the problem to a triviality. Furthermore, we can observe that when the modification rate is low, the GAR training introduces a higher modification rate, as is the case for DeepSeek-Prover. However, if the modification rate is high in the base model, the soft penalty will effectively control it within a reasonable range. These findings prove the improvement of the **GAR**-trained model from another point of view

### D.4 ADDITIONAL ABLATION STUDIES

#### D.4.1 FROZEN FUSER STUDY

To further analyze our generative adversarial training, we conduct this experiment that applies the GAR training only on the prover and keeps the statement fuser untrained. We run three iterations of Frozen fuser GAR training on Goedel-Prover-V2-8B for three iterations. The MiniF2F-Test pass@32 results are demonstrated in Table 7.

From the results, we can see that frozen fuser fails to obtain any performance improvements compared to the base model. This confirms that a static generator is unable to extend the prover's capabilities beyond its initial limits. This proves the necessity of co-evolution for both the problem composer and the prover.

#### D.4.2 SINGLE PROBLEM ENHANCEMENT FUISER

In order to further study the effectiveness of our idea of problem fusing, which enhances the problem difficulty by combining two problems into a single one. We conduct the experiment of replacing the statement fuser with MagicCoder (Wei et al., 2023) that enhances the problem difficulty by restating the provided single problem to make it more advanced. The results are presented in Table 7.

The result indicates that replacing the fusion process with the MagicCoder style problem-evolving strategy results in a 2.46% performance drop compared to the base model. We conclude this degradation to the limited capability of Qwen3-8B to create more advanced problems with only one reference. It may not provide sufficient difficulty gradient for the prover, leading to training failure. In contrast, our fusion approach creates a more effective implicit curriculum by combining distinct concepts from multiple problems, thereby validating the design of the Statement Fuser.

### D.5 CASE STUDY

#### D.5.1 CASE 1:

In this section, we demonstrate an example of how the **GAR** trained model successfully solves the problem while the base models fail. Compared to the base prover, the **GAR** trained model relies less on brute simplification and more on early structural framing, producing shorter, steadier proof traces that remain faithful to the statement and are more likely to pass

Table 7: Additional Ablation Studies

| Method | MiniF2F-Test |
|---|---|
| **Base model** | 77.87% |
| **Frozen Fuser** | 77.87% |
| **MagicCoder Fuser** | 75.41% |
| **GAR trained** | **80.33%** |

the verification. If we dig deeper into the Long CoT of two models, we can find that **GAR**-trained prover writes a shorter and more focused long CoT. While the base model's CoT is longer but less

directed, it cycles among general identities and simplifications without stabilizing on a closing invariant. The result that **GAR**'s trace is both and clearer and more verifiable, which leads to the successful proof.

**GAR-trained model generated result:**

```
<|begin_of_sentence|><|User|>Complete the following Lean 4 code:

```lean4
import Mathlib
import Aesop

set_option maxHeartbeats 0

open BigOperators Real Nat Topology Rat

/-- Consider a prime $p$ of the form $4 t+1$. Show that $a$ is a
↪  primitive root modulo $p$ iff $-a$ is a primitive root modulo $p$. -/
theorem exercise_4_4 {p t: ℕ} (hp0 : p.Prime) (hp1 : p = 4*t + 1)
  (a : ZMod p) :
  IsPrimitiveRoot a p ↔ IsPrimitiveRoot (-a) p := by
```

Before producing the Lean 4 code to formally prove the given theorem,
↪  provide a detailed proof plan outlining the main proof steps and
↪  strategies.
The plan should highlight key ideas, intermediate lemmas, and proof
↪  structures that will guide the construction of the final formal
↪  proof.
<|Assistant|>### Detailed Proof

**Problem Analysis:**
We are given a prime `p` of the form `p = 4 * t + 1` for some natural
↪  number `t`, and an element `a` in the finite field `ZMod p` (where `p`
↪  is the prime number). We need to prove that `a` is a primitive root
↪  modulo `p` if and only if `-a` is a primitive root modulo `p`.

Recall that `IsPrimitiveRoot a p` means that `a` has multiplicative order
↪  `p - 1 = 4 * t` in the multiplicative group `(ZMod p)ˣ`.

**Key Observations:**
1. The multiplicative group `(ZMod p)ˣ` has order `p - 1 = 4 * t`.
2. The negation map `x ↦ -x` is an automorphism of `(ZMod p)ˣ` because
↪  `p` is an odd prime (`p = 4 * t + 1 > 2`, and thus `p` is odd). This
↪  means that `-x` has the same order as `x`.
3. The multiplicative order of `a` and `-a` are the same if and only if `a
↪  = -a`, but this is not the case unless `p` divides `2 * a`, which is
↪  false here because `a` is arbitrary and `p` is prime. However, the
↪  actual condition is that `IsPrimitiveRoot a p` if and only if
↪  `IsPrimitiveRoot (-a) p` because the order of `-a` is the same as the
↪  order of `a` when `p` is of the form `4 * t + 1`.

But actually, we need a better understanding of the multiplicative group
↪  `(ZMod p)ˣ`.

For any odd prime `p`, we have the following properties:
1. The map `x ↦ -x` is an isomorphism on `(ZMod p)ˣ` because `p` is odd.
   - The inverse is `x ↦ -x`, so the map is a permutation of the
   ↪  multiplicative group.
2. For any `x` in `(ZMod p)ˣ`, `x ^ 2` is a square (i.e., `x ^ 2` is a
↪  square in `(ZMod p)ˣ`).
   - To see this, observe that `(a * b)² = a² * b²` and `-1` is a
   ↪  quadratic residue modulo `p`.
```

But for our problem, we don't need these general observations. Instead,
↪  we can use the fact that the order of `-a` is the same as the order
↪  of `a`.

**Why are the orders the same?**

Since `p` is an odd prime (`p = 4 * t + 1 > 2`), the multiplicative group
↪  `(ZMod p)$^x$` is cyclic of order `p - 1 = 4 * t`. Let `g` be a generator
↪  of `(ZMod p)$^x$`.

Then:
1. If `a` is a primitive root (`IsPrimitiveRoot a p` is true), then `a`
↪  has order `4 * t = p - 1`, so `-a` must also have order `4 * t`
↪  because `-a` is a unit in `ZMod p` (`p` is prime, so `-a ≠ 0`).
    - However, this is not correct unless `-1` has order `2` because `-a =
    ↪  (-1) * a`, and the order of `a` is `4 * t`, and the order of `-1`
    ↪  is `2` if `t ≠ 0` (because `(-1)^2 = 1`). Thus, the order of `-a`
    ↪  is `lcm(2, order of a)`.

Alternatively, a better approach is to use the following fact:
- The map `x ↦ -x` is an automorphism of the group `(ZMod p)$^x$` because
↪  `p` is odd (`(-1)^2 = 1` and multiplication is invertible).

**Conclusion:**
Since the map `x ↦ -x` is an automorphism of `(ZMod p)$^x$` (because `p` is
↪  an odd prime), `a` has order `4 * t` if and only if `-a` has order `4
↪  * t`. This is because the automorphism `x ↦ -x` preserves the order
↪  of elements.

Thus, `IsPrimitiveRoot a p` (`order a = 4 * t`) is equivalent to
↪  `IsPrimitiveRoot (-a) p` (`order (-a) = 4 * t`).

Therefore, the result follows.

### Step-by-Step Abstract Plan

1. **Understand the Multiplicative Group**:
    - The multiplicative group `(ZMod p)$^x$` is cyclic of order `p - 1 = 4 *
    ↪  t`.
    - Any element has a multiplicative order dividing `4 * t`.

2. **Understand the Automorphism `x ↦ -x`**:
    - The map `x ↦ -x` is an automorphism of `(ZMod p)$^x$` because `p` is an
    ↪  odd prime (`(-1)^2 = 1`).

3. **Order Preservation**:
    - The order of `-a` is the same as the order of `a`, because `x ↦ -x`
    ↪  is an automorphism.

4. **Conclusion**:
    - `a` is a primitive root (`order a = p - 1`) iff `-a` is a primitive
    ↪  root (`order (-a) = p - 1`).

### Lean 4 `have` Statements

```lean4
theorem exercise_4_4 {p t: ℕ} (hp0 : p.Prime) (hp1 : p = 4*t + 1)
  (a : ZMod p) :
  IsPrimitiveRoot a p ↔ IsPrimitiveRoot (-a) p := by
  have h_main : IsPrimitiveRoot a p ↔ IsPrimitiveRoot (-a) p := by
    sorry
  sorry
```

### Complete Lean 4 Proof

```lean4
theorem exercise_4_4 {p t: ℕ} (hp0 : p.Prime) (hp1 : p = 4*t + 1)
  (a : ZMod p) :
  IsPrimitiveRoot a p ↔ IsPrimitiveRoot (-a) p := by
  have h_main : IsPrimitiveRoot a p ↔ IsPrimitiveRoot (-a) p := by
    haveI : Fact p.Prime := ⟨hp0⟩
    have h₁ : p ≠ 0 := hp0.ne_zero
    have h₂ : p ≠ 1 := hp0.ne_one
    have h₃ : (-a : ZMod p) = -a := rfl
    constructor
    · -- Prove the forward direction: if a is a primitive root, then -a
    ↪  is a primitive root
      intro h
      -- Use the property that the order of -a is the same as the order
      ↪  of a
      have h₄ := h.isUnit
      have h₅ := h.pow_eq_one
      have h₆ := h.pow_ne_one_of_pos_of_lt (by decide : (0 : ℕ) < 1) (by
        rw [← Nat.cast_one]
        simp_all [ZMod.val_one, Nat.cast_add, Nat.cast_mul,
        ↪  Nat.cast_ofNat]
        <;> omega)
      simp_all [IsPrimitiveRoot, ZMod.nat_cast_self, Nat.cast_add,
      ↪  Nat.cast_mul, Nat.cast_ofNat,
        ZMod.nat_cast_self]
      <;>
      (try omega) <;>
      (try simp_all [ZMod.nat_cast_self, Nat.cast_add, Nat.cast_mul,
      ↪  Nat.cast_ofNat,
        ZMod.nat_cast_self]) <;>
      (try
        {
          aesop
        }) <;>
      (try
        {
          simp_all [Units.ext_iff, pow_succ]
          <;>
          ring_nf at * <;>
          omega
        }) <;>
      (try
        {
          norm_num at *
          <;>
          aesop
        }) <;>
      (try
        {
          aesop
        }) <;>
      (try
        {
          simp_all [Units.ext_iff, pow_succ]
          <;>
          ring_nf at * <;>
          omega
        })
      <;>
      aesop
    · -- Prove the reverse direction: if -a is a primitive root, then a
    ↪  is a primitive root
      intro h
```

```
      -- Use the property that the order of -a is the same as the order
      ↪  of a
      have h₄ := h.isUnit
      have h₅ := h.pow_eq_one
      have h₆ := h.pow_ne_one_of_pos_of_lt (by decide : (0 : ℕ) < 1) (by
        rw [← Nat.cast_one]
        simp_all [ZMod.val_one, Nat.cast_add, Nat.cast_mul,
        ↪  Nat.cast_ofNat]
        <;> omega)
      simp_all [IsPrimitiveRoot, ZMod.nat_cast_self, Nat.cast_add,
      ↪  Nat.cast_mul, Nat.cast_ofNat,
        ZMod.nat_cast_self]
      <;>
      (try omega) <;>
      (try simp_all [ZMod.nat_cast_self, Nat.cast_add, Nat.cast_mul,
      ↪  Nat.cast_ofNat,
        ZMod.nat_cast_self]) <;>
      (try
        {
          aesop
        }) <;>
      (try
        {
          simp_all [Units.ext_iff, pow_succ]
          <;>
          ring_nf at * <;>
          omega
        }) <;>
      (try
        {
          norm_num at *
          <;>
          aesop
        }) <;>
      (try
        {
          aesop
        }) <;>
      (try
        {
          simp_all [Units.ext_iff, pow_succ]
          <;>
          ring_nf at * <;>
          omega
        })
      <;>
      aesop
  exact h_main
```
```

**Base model generated result:**

```
<|begin_of_sentence|><|User|>Complete the following Lean 4 code:

```lean4
import Mathlib
import Aesop

set_option maxHeartbeats 0

open BigOperators Real Nat Topology Rat

```

```
/-- Consider a prime $p$ of the form $4 t+1$. Show that $a$ is a
↪  primitive root modulo $p$ iff $-a$ is a primitive root modulo $p$. -/
theorem exercise_4_4 {p t: ℕ} (hp0 : p.Prime) (hp1 : p = 4*t + 1)
  (a : ZMod p) :
  IsPrimitiveRoot a p ↔ IsPrimitiveRoot (-a) p := by
```

Before producing the Lean 4 code to formally prove the given theorem,
↪  provide a detailed proof plan outlining the main proof steps and
↪  strategies.
The plan should highlight key ideas, intermediate lemmas, and proof
↪  structures that will guide the construction of the final formal
↪  proof.
<|Assistant|>### Detailed Proof and Analysis

First, let's recall the definitions and properties we will use:

1. **Primitive Root**: An element `a` is a primitive root modulo `p` if
↪  the multiplicative order of `a` in `(ℤ/pℤ)*` is `p - 1`.
2. **Order of `a`**: The order of `a` modulo `p`, denoted `orderOf a`, is
↪  the smallest positive integer `k` such that `a^k ≡ 1 mod p`.
3. **Multiplicative Order**: The multiplicative order of `a` modulo `p` is
↪  the smallest positive integer `k` such that `a^k ≡ 1 mod p`.

Given `p` is prime and `p = 4*t + 1`, we need to prove that `a` is a
↪  primitive root modulo `p` if and only if `-a` is a primitive root
↪  modulo `p`.

#### Key Observations:
1. The multiplicative group `(ℤ/pℤ)*` has order `p - 1 = 4*t`.
2. The quadratic residues modulo `p` are `{a | ∃ b, b² ≡ a mod p}`. The
↪  number of quadratic residues is `(p - 1)/2 = 2*t` (since `p ≡ 1 mod
↪  4`).
3. The multiplicative order of any `a ∈ (ℤ/pℤ)*` must divide `p - 1 =
↪  4*t`.
4. The multiplicative order of `-a` is the same as the multiplicative
↪  order of `a` because `(-a)^2 = a² ≡ a² mod p`, and the order is
↪  determined by the exponentiation.

#### Proof Sketch:

1. Assume `a` is a primitive root modulo `p`. This means that `a` has
↪  multiplicative order `p - 1 = 4*t` in `(ℤ/pℤ)*`.
   - The order of `-a` is the same as the order of `a`, which is `4*t = p
   ↪  - 1`, because:
     - `(-a)^(4*t) = ((-a)^2)^(2*t) = (a²)^(2*t) ≡ 1 mod p` because `a²`
     ↪  has order `2*t` (since `a` is a primitive root and the exponent
     ↪  `4*t = 2*(2*t)`, but this is not directly relevant).
     - More precisely, `-a` and `a` have the same multiplicative order
     ↪  because `(-a)^(4*t) = ((-a)^2)^(2*t) = (a²)^(2*t) ≡ 1 mod p` (if
     ↪  `a²` has order `2*t`).
   - However, a simpler approach is to observe that `a²` has order `2*t`
   ↪  and thus `(-a)^2 = a²`, so `(-a)^(2*t) = (a²)^t ≡ 1 mod p` is not
   ↪  guaranteed. Instead, we can use that `(-a)^(4*t) = ((-a)^2)^(2*t)`
   ↪  = (a²)^(2*t) ≡ 1 mod p`, but this does not directly help unless we
   ↪  know `a²` has order `2*t`.
   - A better approach is to note that `(-a)^k ≡ 1 mod p` if and only if
   ↪  `a^k ≡ 1 mod p` because:
     - `(-a)^k ≡ 1 mod p` is equivalent to `(-1)^k * a^k ≡ 1 mod p`, so
     ↪  `a^k ≡ (-1)^k mod p`.
     - Since `a` is a primitive root, `a^k ≡ 1 mod p` if and only if `k`
     ↪  is a multiple of `4*t`, because `a` has order `4*t`.

```
    – Thus, `aˆk ≡ (-1)ˆk mod p` would imply `(-1)ˆk ≡ 1 mod p`, i.e.,
    ↪  `(-1)ˆ(4*t) ≡ 1 mod p`, which is true for `p = 4*t + 1`
    ↪  (`(-1)ˆ(4*t + 1) ≡ -1 ≡ 1 mod p`? Wait no: `p = 4*t + 1` and `p
    ↪  ≡ 1 mod 4`, so `(-1)ˆ(4*t + 1) = (-1)ˆ(4*t) * (-1) = 1 * (-1) =
    ↪  -1 ≡ p - 1 mod p`, which is not `1 mod p`. But `p` is prime and
    ↪  `p ≡ 1 mod 4`, so `-1 ≡ p - 1 ≡ 4*t ≡ 0 mod 4`, which is not
    ↪  helpful. This is a mistake in the reasoning.
  – The correct approach is to note that `(-a)ˆ(4*t) = ((-a)ˆ2)ˆ(2*t) =
  ↪  (a²)ˆ(2*t) ≡ 1 mod p` because `a²` has order `2*t`. But `a` is a
  ↪  primitive root, so `aˆ(2*t) ≡ 1 mod p` (`orderOf a = 4*t ≥ 2*t`).
  ↪  This is not sufficient.
  – Alternatively, observe that:
    – The order of `-a` divides `4*t` because the order of `a` divides
    ↪  `4*t`.
    – The order of `-a` does not divide `2*t` unless `a² ≡ 1 mod p`
    ↪  (since `(-a)ˆ2 = a² ≡ 1 mod p` would imply `orderOf (-a) ≤ 2`).
    ↪  But `a` is a primitive root, so `a² ≡ 1 mod p` would imply `2 ≤
    ↪  orderOf a = 4*t`, i.e., `2 ≤ 4*t`, which is true since `t ≥ 1`.
    ↪  But `a² ≡ 1 mod p` is not guaranteed unless `a² ≡ 1 mod p`,
    ↪  which is not necessarily true.
  – A better way is to use the fact that `-a` has the same order as `a`
  ↪  because `p ≡ 1 mod 4`, and `(-a)ˆ2 = a²`.

  However, a simpler and correct approach is to use the following:
  – The multiplicative order of `-a` modulo `p` is the same as the
  ↪  multiplicative order of `a` modulo `p`, because `(-a)ˆ2 = a²` and
  ↪  `a²` has order `2*t` modulo `p`.

  But we can use the following:
  – If `a` is a primitive root modulo `p`, then for any `k`, `aˆk ≡ 1 mod
  ↪  p` if and only if `k` is a multiple of `4*t`.
  – Then `(-a)ˆk ≡ 1 mod p` is equivalent to `aˆk ≡ 1 mod p`, because
  ↪  `(-a)ˆk = (-1)ˆk * aˆk` and `p` is odd (`p ≡ 1 mod 4`). So `(-a)ˆk
  ↪  ≡ 1 mod p` is equivalent to `aˆk ≡ 1 mod p` if `p` is odd.
  – Hence, `orderOf (-a) = orderOf a = 4*t = p - 1`.

  Therefore, `-a` is also a primitive root modulo `p`.

#### Correct Approach:

1. `p` is prime and `p ≡ 1 mod 4`, so `(-a)ˆ2 = a²` modulo `p`.
2. The multiplicative order of `-a` modulo `p` is the same as the
↪  multiplicative order of `a` modulo `p`, because:
  – `(-a)ˆk ≡ 1 mod p` iff `aˆk ≡ 1 mod p` for any `k`, since `(-a)ˆk =
  ↪  (-1)ˆk * aˆk`, and `p` is odd (`p ≡ 1 mod 4`).
  – Alternatively, observe that `orderOf (-a) = orderOf a` because `p ≡
  ↪  1 mod 4` and `a` is a primitive root.
  – In particular, `orderOf (-a) = p - 1 = orderOf a`, because `(-a)ˆ(p
  ↪  - 1) = aˆ(p - 1) ≡ 1 mod p` (Fermat's Little Theorem).

But we need a more precise argument:
1. If `a` is a primitive root, then the multiplicative order of `a` is `p
↪  - 1 = 4*t`.
2. The multiplicative order of `-a` is the same as the multiplicative
↪  order of `a` because `p ≡ 1 mod 4` and `a` is a primitive root.
  – To see this, note that `(-a)ˆ2 = a²`, so `orderOf (-a)` divides
  ↪  `orderOf a²` (which is `2*t`).
  – But `orderOf a²` divides `orderOf a = 4*t` (since `a²ˆ2t = (a²)ˆ2t ≡
  ↪  1 mod p`).
  – Also, `orderOf (-a)` divides `p - 1 = 4*t` because `(-a)ˆ(4*t) =
  ↪  ((-a)ˆ2)ˆ(2*t) = (a²)ˆ(2*t) ≡ 1 mod p`.
  – Since `orderOf (-a)` divides `4*t` and `4*t` is the order of `a`, and
  ↪  `orderOf (-a)` divides `orderOf a²` (which is `2*t`), and `orderOf
  ↪  a²` divides `orderOf a` (which is `4*t`), the only common divisors
  ↪  are `1` and `2*t`.
```

```
  - But `orderOf a = 4*t` and `orderOf a² = 2*t`.
  - Alternatively, because `a` is a primitive root, `a^k ≡ 1 mod p` iff
  ↪  `4*t` divides `k`.
  - Similarly, `(-a)^k ≡ 1 mod p` iff `4*t` divides `k`, because `(-a)^k
  ↪  = a^k` (since `p ≡ 1 mod 4`).
  - Therefore, the order of `-a` is the same as the order of `a`.

But we can simplify this:
- The order of `-a` is the same as the order of `a` because `p ≡ 1 mod 4`
↪  and `a` is a primitive root.

Thus, we have:
1. If `a` is a primitive root, then `-a` is also a primitive root.
2. Similarly, if `-a` is a primitive root, then `a` is also a primitive
↪  root.
```

### Step 1: Abstract Plan

```
1. **Assume `a` is a primitive root modulo `p`**:
   - The multiplicative order of `a` is `p - 1 = 4*t`.
   - We need to show that the multiplicative order of `-a` is `4*t`.
   - Use the fact that `(-a)^k ≡ 1 mod p` iff `a^k ≡ 1 mod p`, because `p
   ↪  ≡ 1 mod 4`.
   - Thus, the order of `-a` is the same as the order of `a`.
   - Therefore, `-a` is also a primitive root.
2. **Assume `-a` is a primitive root modulo `p`**:
   - Similarly, `a` is a primitive root.
   - The reasoning is symmetric.
```

### Step 2: Lean 4 `have` Statements

```lean4
theorem exercise_4_4 {p t: ℕ} (hp0 : p.Prime) (hp1 : p = 4*t + 1)
  (a : ZMod p) :
  IsPrimitiveRoot a p ↔ IsPrimitiveRoot (-a) p := by
  have h_main : IsPrimitiveRoot a p → IsPrimitiveRoot (-a) p := by
    sorry
  have h_converse : IsPrimitiveRoot (-a) p → IsPrimitiveRoot a p := by
    sorry
  have h_final : IsPrimitiveRoot a p ↔ IsPrimitiveRoot (-a) p := by
    sorry
  exact h_final
```

### Explanation:
1. `h_main`: If `a` is a primitive root, then `-a` is also a primitive
↪  root.
2. `h_converse`: If `-a` is a primitive root, then `a` is also a primitive
↪  root.
3. `h_final`: Combine the two directions to get the full biconditional.

### Complete Lean 4 Proof

```lean4
theorem exercise_4_4 {p t: ℕ} (hp0 : p.Prime) (hp1 : p = 4*t + 1)
  (a : ZMod p) :
  IsPrimitiveRoot a p ↔ IsPrimitiveRoot (-a) p := by
  have h_main : IsPrimitiveRoot a p → IsPrimitiveRoot (-a) p := by
    intro h
    haveI := Fact.mk hp0
    have h₁ := h.isUnit
    have h₂ := h.pow_eq_one
    have h₃ := h.pow_ne_one_of_pos_of_lt (by decide : (0 : ℕ) < 1) (by
      simp_all [hp1, Nat.Prime.one_lt]
      <;> omega)
```

```
    have h₄ := h.pow_ne_one_of_pos_of_lt (by decide : (0 : ℕ) < 2) (by
      simp_all [hp1, Nat.Prime.one_lt]
      <;> omega)
    simp_all [ZMod.nat_cast_self, pow_one, pow_two, neg_neg]
    <;>
    (try omega) <;>
    (try simp_all [IsPrimitiveRoot, pow_mul, pow_add, pow_one, pow_two,
    ↪  ZMod.nat_cast_self]) <;>
    (try omega) <;>
    (try
      {
        have h₅ := h.pow_eq_one
        have h₆ := h.isUnit
        simp_all [ZMod.nat_cast_self, pow_one, pow_two, neg_neg]
        <;>
        aesop
      }) <;>
    (try
      {
        have h₅ := h.pow_eq_one
        have h₆ := h.isUnit
        simp_all [ZMod.nat_cast_self, pow_one, pow_two, neg_neg]
        <;>
        aesop
      })
    <;>
    (try
      {
        aesop
      })
    <;>
    (try
      {
        simp_all [ZMod.nat_cast_self, pow_one, pow_two, neg_neg]
        <;>
        aesop
      })
    <;>
    (try
      {
        omega
      })
    <;>
    (try
      {
        aesop
      })
    <;>
    (try
      {
        simp_all [ZMod.nat_cast_self, pow_one, pow_two, neg_neg]
        <;>
        omega
      })

  have h_converse : IsPrimitiveRoot (-a) p → IsPrimitiveRoot a p := by
    intro h
    haveI := Fact.mk hp0
    have h₁ := h.isUnit
    have h₂ := h.pow_eq_one
    have h₃ := h.pow_ne_one_of_pos_of_lt (by decide : (0 : ℕ) < 1) (by
      simp_all [hp1, Nat.Prime.one_lt]
      <;> omega)
    have h₄ := h.pow_ne_one_of_pos_of_lt (by decide : (0 : ℕ) < 2) (by
      simp_all [hp1, Nat.Prime.one_lt]
```

```
      <;> omega)
    simp_all [ZMod.nat_cast_self, pow_one, pow_two, neg_neg]
    <;>
    (try omega) <;>
    (try simp_all [IsPrimitiveRoot, pow_mul, pow_add, pow_one, pow_two,
    ↪  ZMod.nat_cast_self]) <;>
    (try omega) <;>
    (try
      {
        have h₅ := h.pow_eq_one
        have h₆ := h.isUnit
        simp_all [ZMod.nat_cast_self, pow_one, pow_two, neg_neg]
        <;>
        aesop
      }) <;>
    (try
      {
        have h₅ := h.pow_eq_one
        have h₆ := h.isUnit
        simp_all [ZMod.nat_cast_self, pow_one, pow_two, neg_neg]
        <;>
        aesop
      })
    <;>
    (try
      {
        aesop
      })
    <;>
    (try
      {
        simp_all [ZMod.nat_cast_self, pow_one, pow_two, neg_neg]
        <;>
        aesop
      })
    <;>
    (try
      {
        omega
      })
    <;>
    (try
      {
        aesop
      })
    <;>
    (try
      {
        simp_all [ZMod.nat_cast_self, pow_one, pow_two, neg_neg]
        <;>
        omega
      })
    <;>
    aesop

  have h_final : IsPrimitiveRoot a p ↔ IsPrimitiveRoot (-a) p := by
    constructor
    · exact h_main
    · exact h_converse

  exact h_final
```

D.5.2 CASE 2:

In another ProofNet example, **GAR** again succeeds where the base prover fails. This instance more clearly illustrates how **GAR** mitigates useless simplifications. It uses NLs to reach a comprehensive proof plan and effectively conduct the proof in a relatively simple manner. However, the base model conducts a quicker search but a prolonged proof. It expends effort on low-yield algebra and yields lengthy, fragile derivations. These observations highlight that the implicit curriculum in **GAR** fosters more efficient reasoning paths and greater verification robustness.

**GAR-trained model generated result:**

```lean4
<|begin_of_sentence|><|User|>Complete the following Lean 4 code:

```lean4
import Mathlib
import Aesop

set_option maxHeartbeats 0

open BigOperators Real Nat Topology Rat

/-- Prove that $x^4+4x^3+6x^2+2x+1$ is irreducible in $\mathbb{Z}[x]$. -/
theorem exercise_9_4_2c : Irreducible
  (X^4 + 4*X^3 + 6*X^2 + 2*X + 1 : Polynomial ℤ) := by
```

Before producing the Lean 4 code to formally prove the given theorem,
↪   provide a detailed proof plan outlining the main proof steps and
↪   strategies.
The plan should highlight key ideas, intermediate lemmas, and proof
↪   structures that will guide the construction of the final formal
↪   proof.
<|Assistant|>### Detailed Proof

**Problem:** Prove that \( P(x) = x^4 + 4x^3 + 6x^2 + 2x + 1 \) is
↪   irreducible in \(\mathbb{Z}[x]\).

**Approach:**
We will use the Rabinowitz Criterion (also known as the "Rabinowitz
↪   irreducibility test" or "Rabinowitz's Criterion") to prove that \(
↪   P(x) \) is irreducible over \(\mathbb{Z}[x]\).

**Rabinowitz Criterion:**
Let \( P(x) = x^n + a_{n-1}x^{n-1} + \dots + a_0 \) be a polynomial in
↪   \(\mathbb{Z}[x]\). Suppose that there exists a prime \( p \) such
↪   that:
1. \( p \) divides each \( a_i \) for \( 0 \leq i \leq n-1 \).
2. \( p^2 \) does not divide \( a_0 \).
Then \( P(x) \) is irreducible over \(\mathbb{Z}[x]\).

**Application to \( P(x) \):**
Let \( P(x) = x^4 + 4x^3 + 6x^2 + 2x + 1 \).

1. The constant term is \( a_0 = 1 \).
2. The primes dividing all coefficients \( a_0, a_1, a_2, a_3 \) are the
↪   primes dividing \( \gcd(1, 2, 6, 4, 1) = 1 \). So, no primes satisfy
↪   the first condition.
    - Hmm, this is incorrect: actually, all coefficients \( a_0 = 1 \), \(
      ↪   a_1 = 2 \), \( a_2 = 6 \), \( a_3 = 4 \) are divisible by \( 1 \),
      ↪   but the second condition is about \( a_0 = 1 \), not all other
      ↪   coefficients. This suggests that perhaps we need a different
      ↪   approach.
```

```
But the Rabinowitz Criterion can still be applied:
Consider \( p = 2 \).
- \( 2 \) divides \( a_0 = 1 \) (False).
- \( 2 \) divides \( a_1 = 2 \) (True).
- \( 2 \) divides \( a_2 = 6 \) (True).
- \( 2 \) divides \( a_3 = 4 \) (True).
But \( p^2 = 4 \) divides \( a_0 = 1 \) (False).

This does not work. So, we need another prime.

Consider \( p = 3 \):
- \( 3 \) divides \( a_0 = 1 \) (No).
- \( 3 \) divides \( a_1 = 2 \) (No).
\( p \) does not divide any of the coefficients, so this is invalid.

Consider \( p = 5 \):
- \( 5 \) divides \( a_0 = 1 \) (No).
- \( 5 \) divides \( a_1 = 2 \) (No).
\( p \) does not divide any of the coefficients, so this is invalid.

**Alternative Approach:**
Let's consider the polynomial \( Q(x) = x^4 + 4x^3 + 6x^2 + 2x + 1 \). We
↪ can attempt to factor it or check for possible roots.

But \( Q(0) = 1 \), \( Q(1) = 1 + 4 + 6 + 2 + 1 = 14 \), and \( Q(-1) = 1
↪ - 4 + 6 - 2 + 1 = 2 \). No simple rational roots.

Alternatively, perhaps we can factor \( Q(x) \) into quadratics.

Assume \( Q(x) = (x^2 + a x + b)(x^2 + c x + d) \).

Expanding gives:
\[ x^4 + (a + c)x^3 + (ac + b + d)x^2 + (ad + bc)x + bd = Q(x). \]

Thus, we get the system:
1. \( a + c = 4 \)
2. \( ac + b + d = 6 \)
3. \( ad + bc = 2 \)
4. \( bd = 1 \)

From \( bd = 1 \), since \( b, d \in \mathbb{Z} \), we have the following
↪ cases:
1. \( b = 1, d = 1 \), or
2. \( b = -1, d = -1 \).

**Case 1: \( b = d = 1 \)**
From (1): \( a + c = 4 \)
From (2): \( ac + 2 = 6 \implies ac = 4 \)
From (3): \( a + c = 4 \), \( ac = 4 \)
But \( a, c \) are roots of \( t^2 - 4t + 4 = 0 \), i.e., \( t = 2 \). So
↪ \( a = c = 2 \).

This satisfies all equations:
- \( a + c = 4 \)
- \( ac = 4 \)
- \( ad + bc = 2 \cdot 1 + 2 \cdot 1 = 4 \neq 2 \) (Does not hold).

Wait, this is incorrect. The problem is in (3):
\( ad + bc = a \cdot 1 + c \cdot 1 = (a + c) = 4 \neq 2 \), which
↪ contradicts the third condition.

Hence, this case is invalid.

**Case 2: \( b = d = -1 \)**
From (1): \( a + c = 4 \)
```

```
From (2): \( ac - 2 = 6 \implies ac = 8 \)
From (3): \( -a + -c = 2 \implies a + c = -2 \), but this contradicts \(
↪  a + c = 4 \).

This case is also invalid.

**Another Approach: Eisenstein's Criterion**
This polynomial is not directly suited for Eisenstein's Criterion,
↪  because replacing \( x \) with \( x + 1 \) gives:
\[ (x + 1)^4 + 4(x + 1)^3 + 6(x + 1)^2 + 2(x + 1) + 1 \]
\[ = x^4 + 4x^3 + 6x^2 + 4x + 1 + 4x^3 + 12x^2 + 12x + 4 + 6x^2 + 12x + 6
↪  + 2x + 2 + 1 \]
\[ = x^4 + (4x^3 + 4x^3) + (6x^2 + 12x^2 + 6x^2) + (4x + 12x + 12x + 2x)
↪  + (1 + 4 + 6 + 2 + 1) \]
\[ = x^4 + 8x^3 + 24x^2 + 30x + 14 \]
But we can try another substitution.

Alternatively, observe that:
\[ Q(x) = x^4 + 4x^3 + 6x^2 + 2x + 1 \]
\[ = (x^2 + 2x)^2 + 2x^2 + 2x + 1 \]
But this seems not helpful.

However, a better approach is to note that:
\[ Q(x) = (x^2 + 2x + 1)^2 - (2x^2 + 1) \]
But:
\[ Q(x) = x^4 + 4x^3 + 6x^2 + 2x + 1 \]
But:
\[ (x^2 + 2x + 1)^2 = x^4 + 4x^3 + 6x^2 + 4x + 1 \]
\[ Q(x) = (x^2 + 2x + 1)^2 - (2x + 2) \]
But \( (x^2 + 2x + 1)^2 - (2x + 2) = x^4 + 4x^3 + 6x^2 + 4x + 1 - 2x - 2
↪  = x^4 + 4x^3 + 6x^2 + 2x - 1 \neq Q(x) \).

This substitution is incorrect.

**Conclusion from the failed attempts:**
It seems difficult to factor \( Q(x) \) in \(\mathbb{Z}[x]\) by simple
↪  methods.

But we can try a more general approach using the Cohn's Irreducibility
↪  Criterion or Ritt's Criterion, but these are not straightforward.

However, we can use the Rational Root Test (or simply checking small
↪  primes):
Check for \( x = \pm 1, \pm p \) for primes \( p \):
- \( Q(1) = 1 + 4 + 6 + 2 + 1 = 14 \)
- \( Q(-1) = 1 - 4 + 6 - 2 + 1 = 2 \)
- \( Q(2) = 16 + 32 + 24 + 4 + 1 = 77 \)
- \( Q(-2) = 16 - 32 + 24 - 4 + 1 = 5 \)

Since \( Q(x) \) has no rational roots, it is irreducible over
↪  \(\mathbb{Z}\).

But this seems incorrect, as \( Q(x) \) is reducible for \( x = 0 \) and
↪  \( x = -1 \), but we are not using this approach.

Alternatively, we can use the **Schinzel's criterion** or **Rabinowitz
↪  Criterion** again:
Consider \( Q(x + 1) = (x + 1)^4 + 4(x + 1)^3 + 6(x + 1)^2 + 2(x + 1) + 1
↪  \)
\[ = x^4 + 4x^3 + 6x^2 + 4x + 1 + 4x^3 + 12x^2 + 12x + 4 + 6x^2 + 12x + 6
↪  + 2x + 2 + 1 \]
\[ = x^4 + (4x^3 + 4x^3) + (6x^2 + 12x^2 + 6x^2) + (4x + 12x + 12x + 2x)
↪  + (1 + 4 + 6 + 2 + 1) \]
\[ = x^4 + 8x^3 + 24x^2 + 30x + 14 \]
```

```
Hmm, no immediate factorization here either.

**Final Attempt:**
We use the fact that if \( Q(x) \) is irreducible over \(\mathbb{Z}\),
↪  then it is irreducible over \(\mathbb{Q}\).

Alternatively, we can directly check that \( Q(x) \) has no factors in
↪  \(\mathbb{Z}[x]\) other than \( Q(x) \) itself and constants.

We can perform a complete check:

Suppose \( Q(x) = A(x) B(x) \) is a factorization into non-constant
↪  polynomials.

We can write:
\[ A(x) = x^4 + a_1 x^3 + a_2 x^2 + a_3 x + a_4 \]
\[ B(x) = 1 \]
But \( B(x) \) is a unit and \( \deg(B(x)) = 0 \), so this is trivial.

Alternatively, assume \( A(x) \) has degree \( 1 \) or \( 2 \), etc.

But this is tedious and unlikely to work.

Given that the problem is from an older source, we can infer that the
↪  polynomial \( Q(x) \) is irreducible.

But in Lean, we can proceed by proving that \( Q(x) \) has no non-trivial
↪  factors, by checking that for every pair of integers \( (a, b) \)
↪  such that \( b \neq \pm 1 \), we cannot write \( Q(x) \) as a product
↪  of two non-constant polynomials.

Alternatively, we can use Lean's libraries to prove irreducibility by
↪  contradiction:

**Proof Sketch:**
1. Assume \( Q(x) = A(x) B(x) \).
2. Look at the constant term.
   The constant term is \( 1 \), so possible constant factors are \( \pm
   ↪  1 \).
3. So either \( A(x) \) or \( B(x) \) is a constant, say \( A(x) = \pm 1
↪  \).
4. But then \( Q(x) \) would have to be constant, which is false.
   So, contradiction.

But Lean may not directly have lemmas for this.

A better approach is to use that for a monic polynomial with integer
↪  coefficients and \( \pm 1 \) constant term, it is irreducible if it
↪  has no non-trivial factors.

But in Lean, we can prove that no non-trivial factors exist.

However, checking all possible factors is impractical.

Alternatively, we can use the fact that \( Q(x) \) is a cyclotomic-like
↪  polynomial or a Schur polynomial.

But since this seems too involved, we can use Lean's **irreducibility
↪  testing** (but that's not part of standard libraries).

**Alternative Step:**
We can instead show that \( Q(x) \) is not a perfect square in \(
↪  \mathbb{Z}[x] \).

But this is non-trivial.
```

```
Alternatively, we can use **Mignotte's bound** to check that no
↪ non-trivial factors exist.

But given that \( Q(x) \) is not a known factorizable polynomial, we can
↪ simply use the fact that it is not reducible and is of degree \( 4
↪ \), so it must be irreducible if it has no roots in \( \mathbb{Z} \).

Since \( Q(x) \) is strictly increasing for \( x \geq 0 \) and \( Q(-1) =
↪ 2 \), \( Q(0) = 1 \), and \( Q(1) = 14 \), there are no integer
↪ roots.

But Lean can confirm this by checking values.

But instead, we can use:

**Lemmas:**
1. If \( Q(x) \) is reducible in \( \mathbb{Z}[x] \), then it has a
↪ linear or quadratic factor.
2. A reducible polynomial must have a factor with \( \deg \leq \lfloor
↪ \frac{deg(Q)}{2} \rfloor = 2 \).
3. The possible quadratics are \( x^2 + a x + 1 \) or similar, but none
↪ divide \( Q(x) \).

But checking this is tedious.

Instead, we can appeal to a known fact that \( x^4 + x^3 + x^2 + x + 1 \)
↪ is irreducible, and substituting \( x + 1 \) gives:
\[ (x + 1)^4 + 4(x + 1)^3 + 6(x + 1)^2 + 2(x + 1) + 1 \]
\[ = x^4 + 4x^3 + 6x^2 + 4x + 1 + 4x^3 + 12x^2 + 12x + 4 + 6x^2 + 12x + 6
↪ + 2x + 2 + 1 \]
\[ = x^4 + 8x^3 + 24x^2 + 30x + 14 \]
This is the same as above.

Alternatively, check that \( x^4 + 4x^3 + 6x^2 + 2x + 1 \) is irreducible
↪ using:

**Useful Lemma:** The polynomial \( x^4 + a x^3 + b x^2 + c x + 1 \) is
↪ irreducible in \( \mathbb{Z}[x] \) if \( a^2 < 4b \) or \( b^2 < 4c
↪ \), etc.

But this is not directly helpful.

**Lean-Applicable Approach:**
To prove irreducibility in Lean, we can directly use:
- The **reduction modulo primes** approach to eliminate possible
↪ factorizations.

Alternatively, we can use the fact that \( Q(x) \) is a cyclotomic-like
↪ polynomial or a Cohn polynomial.

But given that the polynomial is of degree \( 4 \) and no obvious
↪ factorization exists, we can proceed by contradiction.

### Step-by-Step Abstract Plan

1. **Assume Reducibility**: Suppose \( Q(x) \) is reducible in \(
↪ \mathbb{Z}[x] \). Then, \( Q(x) = A(x) B(x) \) with \( \deg(A(x))
↪ \geq 1 \), \( \deg(B(x)) \geq 1 \), and \( \deg(A(x)) + \deg(B(x)) =
↪ 4 \).

2. **Analyze Constant Terms**: Since \( Q(0) = 1 \) and \( Q(x) \) has no
↪ rational roots, no constant factors can exist unless one of \( A(x)
↪ \) or \( B(x) \) is \( \pm 1 \).
```

3. **Check for Linear Factors**: If \( \deg(A(x)) = 1 \), say \( A(x) = x + a \), then \( Q(x) = (x + a) B(x) \). Expand to get coefficients and contradict.

4. **Check for Quadratic Factors**: If \( \deg(A(x)) = 2 \), say \( A(x) = x^2 + a x + b \), analyze to find contradictions.

5. **Contradiction**: Since all possible factorizations lead to contradictions, \( Q(x) \) must be irreducible.

### Lean 4 Abstract Plan with `have` Statements

```lean4
theorem exercise_9_4_2c : Irreducible
  (X^4 + 4*X^3 + 6*X^2 + 2*X + 1 : Polynomial ℤ) := by
  have h_main : Irreducible (X^4 + 4*X^3 + 6*X^2 + 2*X + 1 : Polynomial
   ↪  ℤ) := by
    sorry
  sorry
```

### Complete Lean 4 Proof

```lean4
theorem exercise_9_4_2c : Irreducible
  (X^4 + 4*X^3 + 6*X^2 + 2*X + 1 : Polynomial ℤ) := by
  have h_main : Irreducible (X^4 + 4*X^3 + 6*X^2 + 2*X + 1 : Polynomial
   ↪  ℤ) := by
    apply?
  apply h_main
```

**Base model generated result:**

<|begin␣of␣sentence|><|User|>Complete the following Lean 4 code:

```lean4
import Mathlib
import Aesop

set_option maxHeartbeats 0

open BigOperators Real Nat Topology Rat

/-- Prove that $x^4+4x^3+6x^2+2x+1$ is irreducible in $\mathbb{Z}[x]$. -/
theorem exercise_9_4_2c : Irreducible
  (X^4 + 4*X^3 + 6*X^2 + 2*X + 1 : Polynomial ℤ) := by
```

Before producing the Lean 4 code to formally prove the given theorem,
↪  provide a detailed proof plan outlining the main proof steps and
↪  strategies.
The plan should highlight key ideas, intermediate lemmas, and proof
↪  structures that will guide the construction of the final formal
↪  proof.
<|Assistant|>### Detailed Proof and Analysis

We need to prove that the polynomial \( P(x) = x^4 + 4x^3 + 6x^2 + 2x + 1
↪  \) is irreducible over \(\mathbb{Z}[x]\).

#### Step 1: Understand the Polynomial

```
First, observe that \( P(x) \) is palindromic. A palindromic polynomial
↪  \( P(x) \) can be factored as \( P(x) = (x^2 + a x + b)(x^2 + c x +
↪  d) \) if and only if \( P(x) \) can be written as a product of two
↪  quadratic polynomials.
```

#### Step 2: Try to Factor \( P(x) \)
```
Assume that \( P(x) \) factors into two non-constant polynomials with
↪  integer coefficients:
\[ P(x) = (x^2 + a x + b)(x^2 + c x + d). \]

Expanding the right-hand side gives:
\[ x^4 + (a + c)x^3 + (ac + b + d)x^2 + (ad + bc)x + bd. \]

By comparing coefficients, we get the system:
1. \( a + c = 4 \),
2. \( ac + b + d = 6 \),
3. \( ad + bc = 2 \),
4. \( bd = 1 \).
```

#### Step 3: Solve the System
```
From \( bd = 1 \), the possible integer pairs \((b, d)\) are:
\[ (1, 1), (-1, -1). \]

**Case 1:** \( b = 1 \), \( d = 1 \).
Substitute into the other equations:
1. \( a + c = 4 \),
2. \( ac + 1 + 1 = 6 \Rightarrow ac = 4 \),
3. \( a + c = 4 \),
4. \( ad + bc = a + c = 4 \), but this is already satisfied.

But from \( a + c = 4 \) and \( ac = 4 \), the quadratic equation for \(
↪  a \) is:
\[ t^2 - 4t + 4 = 0 \Rightarrow (t - 2)^2 = 0 \Rightarrow t = 2. \]
Thus \( a = c = 2 \).

Check the last equation: \( ad + bc = 2 \cdot 1 + 2 \cdot 1 = 4 \neq 2
↪  \). This is a contradiction, so this case is invalid.

**Case 2:** \( b = -1 \), \( d = -1 \).
Substitute into the other equations:
1. \( a + c = 4 \),
2. \( ac - 1 - 1 = 6 \Rightarrow ac = 8 \),
3. \( ad + bc = -a - c = 2 \),
4. \( bd = 1 \).

But from \( a + c = 4 \) and \( ac = 8 \), the quadratic equation for \(
↪  a \) is:
\[ t^2 - 4t + 8 = 0, \]
which has discriminant \( D = 16 - 32 = -16 < 0 \). Thus, there are no
↪  real (and hence no integer) roots, so this case is invalid.
```

#### Step 4: Check Other Possible Factorizations
```
The above exhaustive search shows that \( P(x) \) does not factor into
↪  two non-constant polynomials with integer coefficients. Thus, \( P(x)
↪  \) is irreducible.
```

#### Verification of the Irreducibility
```
We have checked all possible factorizations of \( P(x) \) into quadratics
↪  and found that none satisfy the coefficient conditions. Therefore, \(
↪  P(x) \) is irreducible.
```

### Abstract Plan

```
1. **Assume Factorization**: Suppose \( P(x) = (x^2 + a x + b)(x^2 + c x
↪  + d) \) for integers \( a, b, c, d \).
```

2. **Compare Coefficients**: Derive the system of equations for \( a, b,
↪ c, d \).
   - \( a + c = 4 \),
   - \( ac + b + d = 6 \),
   - \( ad + bc = 2 \),
   - \( bd = 1 \).
3. **Solve \( bd = 1 \)**: Consider the pairs \((b, d)\) as \((1, 1)\)
↪ and \((-1, -1)\).
4. **Case \( b = 1 \), \( d = 1 \)**:
   - Derive \( a + c = 4 \) and \( ac = 4 \).
   - Solve for \( a \) and \( c \) to find \( a = c = 2 \).
   - But this leads to \( ad + bc = 4 \neq 2 \), a contradiction.
5. **Case \( b = -1 \), \( d = -1 \)**:
   - Derive \( a + c = 4 \) and \( ac = 8 \).
   - Show that no real roots exist for \( a \), so this case is invalid.
6. **Conclusion**: No valid factorization exists, so \( P(x) \) is
↪ irreducible.

### Lean 4 `have` Statements

```lean4
theorem exercise_9_4_2c : Irreducible
  (X^4 + 4*X^3 + 6*X^2 + 2*X + 1 : Polynomial ℤ) := by
  have h_main : Irreducible (X^4 + 4*X^3 + 6*X^2 + 2*X + 1 : Polynomial
  ↪  ℤ) := by sorry
  sorry
```

### Complete Lean 4 Proof

```lean4
theorem exercise_9_4_2c : Irreducible
  (X^4 + 4*X^3 + 6*X^2 + 2*X + 1 : Polynomial ℤ) := by
  have h_main : Irreducible (X^4 + 4*X^3 + 6*X^2 + 2*X + 1 : Polynomial
  ↪  ℤ) := by
    -- Assume the polynomial factors into two non-constant polynomials
    ↪  with integer coefficients
    refine' (prime_of_irreducible (show Prime (X^4 + 4*X^3 + 6*X^2 + 2*X
    ↪  + 1 : Polynomial ℤ) by
      -- Use the fact that the polynomial is prime to show it is
      ↪  irreducible
      apply prime_C_mul_X_pow_add_C_of_prime
      <;> norm_num
      <;>
        decide))
    <;>
    (try norm_num) <;>
    (try decide) <;>
    (try
      {
        intro h
        -- If the polynomial factors, it must factor into quadratics
        rcases h with ⟨p, q, hp, hq, hpq⟩
        -- Analyze the possible factorizations
        have h₁ := congr_arg (fun p => eval 0 p) hpq
        have h₂ := congr_arg (fun p => eval 1 p) hpq
        have h₃ := congr_arg (fun p => eval (-1) p) hpq
        have h₄ := congr_arg (fun p => eval 2 p) hpq
        have h₅ := congr_arg (fun p => eval (-2) p) hpq
        simp at h₁ h₂ h₃ h₄ h₅
        -- Check the possible evaluations to find a contradiction
        norm_num [eval_add, eval_mul, eval_pow, eval_C, eval_X, eval_pow,
        ↪  eval_mul, eval_add, eval_sub] at h₁ h₂ h₃ h₄ h₅
        <;>
        (try { contradiction }) <;>
```

```
(try {
  have h₆ := p.sub_dvd_eval_sub 0 1
  have h₇ := q.sub_dvd_eval_sub 0 1
  have h₈ := p.sub_dvd_eval_sub 0 (-1)
  have h₉ := q.sub_dvd_eval_sub 0 (-1)
  have h₁₀ := p.sub_dvd_eval_sub 0 2
  have h₁₁ := q.sub_dvd_eval_sub 0 2
  have h₁₂ := p.sub_dvd_eval_sub 0 (-2)
  have h₁₃ := q.sub_dvd_eval_sub 0 (-2)
  simp [eval_add, eval_mul, eval_pow, eval_C, eval_X] at h₆ h₇ h₈
  ↪  h₉ h₁₀ h₁₁ h₁₂ h₁₃
  <;>
  (try omega) <;>
  (try {
    have h₁₄ : p.natDegree ≤ 2 := by
      contrapose! h₆
      simp_all [Polynomial.eval_add, Polynomial.eval_mul,
      ↪  Polynomial.eval_pow, Polynomial.eval_C,
      ↪  Polynomial.eval_X]
      <;>
      (try omega) <;>
      (try nlinarith)
    have h₁₅ : q.natDegree ≤ 2 := by
      contrapose! h₇
      simp_all [Polynomial.eval_add, Polynomial.eval_mul,
      ↪  Polynomial.eval_pow, Polynomial.eval_C,
      ↪  Polynomial.eval_X]
      <;>
      (try omega) <;>
      (try nlinarith)
    have h₁₆ : p.natDegree ≤ 2 := by
      contrapose! h₈
      simp_all [Polynomial.eval_add, Polynomial.eval_mul,
      ↪  Polynomial.eval_pow, Polynomial.eval_C,
      ↪  Polynomial.eval_X]
      <;>
      (try omega) <;>
      (try nlinarith)
    have h₁₇ : q.natDegree ≤ 2 := by
      contrapose! h₉
      simp_all [Polynomial.eval_add, Polynomial.eval_mul,
      ↪  Polynomial.eval_pow, Polynomial.eval_C,
      ↪  Polynomial.eval_X]
      <;>
      (try omega) <;>
      (try nlinarith)
    have h₁₈ : p.natDegree ≤ 2 := by
      contrapose! h₁₀
      simp_all [Polynomial.eval_add, Polynomial.eval_mul,
      ↪  Polynomial.eval_pow, Polynomial.eval_C,
      ↪  Polynomial.eval_X]
      <;>
      (try omega) <;>
      (try nlinarith)
    have h₁₉ : q.natDegree ≤ 2 := by
      contrapose! h₁₁
      simp_all [Polynomial.eval_add, Polynomial.eval_mul,
      ↪  Polynomial.eval_pow, Polynomial.eval_C,
      ↪  Polynomial.eval_X]
      <;>
      (try omega) <;>
      (try nlinarith)
    have h₂₀ : p.natDegree ≤ 2 := by
      contrapose! h₁₂
```

```
        simp_all [Polynomial.eval_add, Polynomial.eval_mul,
        ↪  Polynomial.eval_pow, Polynomial.eval_C,
        ↪  Polynomial.eval_X]
        <;>
        (try omega) <;>
        (try nlinarith)
      have h₂₁ : q.natDegree ≤ 2 := by
        contrapose! h₁₃
        simp_all [Polynomial.eval_add, Polynomial.eval_mul,
        ↪  Polynomial.eval_pow, Polynomial.eval_C,
        ↪  Polynomial.eval_X]
        <;>
        (try omega) <;>
        (try nlinarith)
      -- Now we know the degree of p and q is at most 2
      have h₂₂ : p.natDegree ≤ 2 := by linarith
      have h₂₃ : q.natDegree ≤ 2 := by linarith
      have h₂₄ : p = C (p.coeff 0) ∨ p = C (p.coeff 0) * X ∨ p = C
      ↪  (p.coeff 0) * X ^ 2 := by
        have h₂₅ := p.natDegree_le_of_degree_le h₂₂
        have h₂₆ := p.natDegree_le_of_degree_le h₂₂
        have h₂₇ := p.natDegree_le_of_degree_le h₂₂
        cases' lt_or_eq_of_le (natDegree_le_of_degree_le h₂₂) with
        ↪  h₂₈ h₂₈ <;>
          simp_all [Polynomial.natDegree_le_iff_degree_le,
          ↪  Polynomial.degree_eq_natDegree]
          <;>
          (try omega) <;>
          (try aesop) <;>
          (try {
            aesop
          }) <;>
          (try {
            aesop
          }) <;>
          (try {
            aesop
          })
      have h₂₅ : q = C (q.coeff 0) ∨ q = C (q.coeff 0) * X ∨ q = C
      ↪  (q.coeff 0) * X ^ 2 := by
        have h₂₆ := q.natDegree_le_of_degree_le h₂₃
        have h₂₇ := q.natDegree_le_of_degree_le h₂₃
        have h₂₈ := q.natDegree_le_of_degree_le h₂₃
        cases' lt_or_eq_of_le (natDegree_le_of_degree_le h₂₃) with
        ↪  h₂₉ h₂₉ <;>
          simp_all [Polynomial.natDegree_le_iff_degree_le,
          ↪  Polynomial.degree_eq_natDegree]
          <;>
          (try omega) <;>
          (try aesop) <;>
          (try {
            aesop
          }) <;>
          (try {
            aesop
          }) <;>
          (try {
            aesop
          })
      rcases h₂₄ with (h₂₅ | h₂₅ | h₂₅) <;> rcases h₂₅ with (h₂₆ | h₂₆
      ↪  | h₂₆) <;> simp_all [Polynomial.eval_add,
      ↪  Polynomial.eval_mul, Polynomial.eval_pow,
      ↪  Polynomial.eval_C, Polynomial.eval_X]
      <;>
      (try omega) <;>
```

```
          (try nlinarith) <;>
          (try {
            ring_nf at *
            <;>
            omega
          }) <;>
          (try {
            norm_num at *
            <;>
            aesop
          })
        })
      })
    })
  <;>
  aesop
 exact h_main
```