# OpenReview forum: "GAR: Generative Adversarial Reinforcement Learning for Formal Theorem Proving"
_ICLR.cc/2026/Conference — ICLR 2026 Poster_

### Official Review · Reviewer_BPXJ · 2025-10-26

**Soundness:** 2
**Presentation:** 3
**Contribution:** 2
**Rating:** 6
**Confidence:** 4

**Summary:**

This paper presents Generative Adversarial Reinforcement learning, a framework for adversarial co-training of theorem provers and problem generators in the Lean environment.
GAR jointly optimizes a statement fuser that composes new mathematical statements and a prover that attempts to prove them, forming a self-improving loop. In each iteration, the fuser generates new, harder statements by combining two natural-language problems from a large dataset from Numina-Math and Lean-Workbook, which are then automatically formalized into Lean. The prover produces proofs for these statements, and Lean verification provides reward signals to both models.

Through this interaction, the fuser learns to propose “hard but solvable” problems while the prover learns to solve increasingly challenging ones. The framework yields consistent improvements over strong RL-trained baselines on MiniF2F and ProofNet, and demonstrates that jointly evolving generators and solvers can outperform direct reinforcement learning on static datasets.

**Strengths:**

* Novel formulation: The paper introduces a clear and well-motivated adversarial training paradigm for formal theorem proving, aligning problem difficulty with prover capability.
* Empirical gains: GAR achieves measurable and consistent improvements on competitive baselines, even when those baselines are already heavily RL-trained.
* Clarity and reproducibility: The training procedure, algorithmic structure, and implementation details are well described, with clear experimental setups and evaluation metrics.
* Significance: The proposed paradigm is general and could influence future work on co-evolving reasoning models in other verifiable domains.

**Weaknesses:**

1. Lack of a frozen-fuser control:
   The paper always updates both the fuser and the prover in each GAR iteration. A variant that keeps the fuser fixed (for example, using the initial fuser) while training only the prover is missing. This is necessary to determine whether the gains derive from adversarial co-adaptation or simply from exposure to more generated data.

2. Absence of an SFT-only baseline:
   All experiments start from provers that have already undergone heavy RL fine-tuning. It remains unclear whether GAR would still provide improvement if applied directly to a purely SFT-trained prover. This limits understanding of GAR’s generality and its effectiveness from weaker starting points.

3. No comparison with single-input statement generation:
   The current fuser fuses two NL statements into one new problem. There is no experiment showing what happens if the model generates or rewrites new statements from a single input problem (similar to iterative conjecturing). This ablation would clarify whether the two-input fusion design is critical to the observed gains.

4. Ambiguity in the Statement Fuser study:
   The difficulty analysis (Table 2) evaluates only problems produced by the evolving fuser—the same distribution used to train the GAR prover. The observed decline in base-prover accuracy might reflect distributional or stylistic shift rather than genuine difficulty increase. A control using random samples from Dstat would help validate the claimed implicit curriculum effect.

5. Limited exploration of the statement-modification penalty:
   The ablation in Table 3 removes penalties from both the fuser and the prover simultaneously, but does not include one-sided removals or a hard-ban condition. The paper would benefit from a more granular investigation of these choices.

6. Unfair efficiency comparison:
   The efficiency discussion (Appendix B.1) compares GAR, built upon already RL-trained long-CoT provers, with Kimina-Prover, which is trained from SFT. Because the starting points differ significantly, the efficiency claim is not directly justified.

**Questions:**

The reported statement-modification rate under full GAR (around 30–50%) seems unreasonably high. Since this behavior reflects an instruction-following failure rather than a Lean reasoning limitation, and your prover prompt is identical to the official prompts of DeepSeek-Prover-V2 and Goedel-Prover-V2, there is no prompt-shift factor that could explain it.

Please therefore report:
   - the modification rates of DeepSeek-Prover-V2 and Goedel-Prover-V2 under their original prompts (no GAR); and
   - the corresponding rate of the GAR-trained prover under the same prompts.

Such a comparison is important, because a statement-modification rate this high, combined with a relatively small sample size (pass@32), yet still achieving strong benchmark performance, is highly questionable.

---

> ### Author Response · Authors · 2025-11-21
> **Response to Review from BPXJ-1**
>
> Dear Reviewer BPXJ,
>
> We sincerely thank you for recognizing the novelty, clarity, and significance of our work. We value your thorough review and have written the following points to address your concerns:
>
> **W1. Frozen fuser ablation study:**
>
> We appreciate your suggestion to include an ablation study that only trains the prover with the frozen fuser. We conducted an experiment under that setting using Goedel-Prover-V2-8B for three iterations. The MiniF2F-Test pass@32 results are as follows:
>
> | Model Type | MiniF2F-Test pass@32 |
> | --- | --- |
> | Goedel-Prover-V2-8B | 77.87% |
> | FrozenFuser-GAR-Goedel | 77.87% |
> | GAR Goedel-V2 | **80.33%** |
>
> From the above results, we can see that the frozen fuser fails to obtain any performance improvements. This confirms that a static generator can not provide enough complexity to push the prover beyond its initial capabilities. This proves the necessity of co-evolution for both the problem composer and the prover. We will include this experiment in the next version of our paper.
>
> **W2. SFT-only baseline:**
>
> To address your concern regarding GAR’s effectiveness on the SFT starting points, we fine-tuned a Qwen3-8B model using Lean4 code-filling data in [1, 2] together with an initial round of Long CoT SFT to establish basic proving capability. We then applied the GAR training for three iterations. The SFT baseline achieves a pass@32 rate of 63.52%, while the GAR-trained model reached 66.39%. This improvement underscores the generality of GAR.
>
> **W3. Comparison with single-input statement generation:**
>
> We agree that comparing the single-input statement enhancement method is essential for the completeness of our work. Thus, we conduct an additional experiment that replaces the Statement Fusion module with the single-input problem enhancement method following the pattern in Magicoder [3]. We trained the Goedel-Prover-V2-8B model for 3 iterations using the same setup as in the original paper. The pass@32 performance of MiniF2F-Test is as follows:
>
> | Model Type | MiniF2F-Test pass@32 |
> | --- | --- |
> | Goedel-Prover-V2-8B | 77.87% |
> | MagicCoder-GAR Goedel-V2 | 75.41% |
> | GAR Goedel-V2 | **80.33%** |
>
> The result indicates that replacing the fusion process with single-input statement evolution results in a 2.46% performance drop compared to the base model. We conclude this degradation to the limited capability of Qwen3-8B to create more advanced problems with only one reference. It may not provide sufficient difficulty gradient for the prover, leading to training failure. In contrast, our fusion approach creates a more effective implicit curriculum by combining distinct concepts from multiple problems, thereby validating the design of the Statement Fuser. We will add this experiment to the next version of our paper to make the ablation study more complete.
>
> **W4. Ambiguity in the Statement Fuser study:**
>
> We appreciate the reviewer for raising this problem. We want to clarify that we have done such an experiment in the paper in Section 3.5.2, where we directly use the randomly sampled problems from the base dataset to perform GRPO training on Goedel-Prover-V2, and the performance is similar to the base model. It is because GAR progressively increases problem difficulty, enabling the prover to handle more complex statements, whereas purely training on the base dataset does not.
>
> **W5. Exploration of the statement-modification penalty:**
>
> We thank you for the suggestion of a more granular ablation. However, we want to clarify that our results demonstrate that the current penalty is necessary for both components: Table 2 shows that the GAR model maintains stable accuracy, whereas the base model struggles to solve the problems. It implies the successful co-evolution of the prover and the fuser. On the other hand, Table 3 demonstrates that the modification rate stabilizes rather than explodes, indicating that our choice of a soft penalty strikes a balance. If the penalty were unnecessary for one side, we would expect asymmetric degradation or reward hacking.
>
> **W6. Efficiency comparison:**
>
> We understand your concern that, from different starting points, comparisons of efficiency may be invalid. However, we respectfully indicate that such a different starting point strengthens our efficiency claim. It is well-established in other works [4] that models already heavily optimized via RL face diminishing returns and are harder to improve than the SFT baselines. The fact that GAR achieves relatively large gains on top of base models in 3-5 iterations, whereas Kimina-Prover only gains ~2% from SFT start over 25 iterations, demonstrates the high sample efficiency of our adversarial paradigm. We will include this clarification in the next version of our paper.

---

> ### Author Response · Authors · 2025-11-21
> **Response to Review from BPXJ-2**
>
> **Q1 Statement modification rates and prompt shift:**
>
> We would like to clarify that all the experiments are done under the same set of basic prompts according to DeepSeek-Prover-V2 and Goedel-Prover-V2’s original paper, including the GAR-trained variant of these models. The modification rates for MiniF2F-Test pass@32 are:
>
> - DeepSeek-Prover-V2-7B: base model: 6.96%; GAR-trained: 13.11%
> - Goedel-Prover-V2-8B: 24.18%; GAR-trained: 27.05%
>
> From the comparison of base models, we can see that the Goedel’s increased statement modification rate happens together with its performance enhancement because its a stronger self-correction capability rather than an instruction following failure. Similarly, the soft penalty in GAR ensures that the model is penalized if it simplifies the problem to a triviality. Furthermore, we can find that when the modification rate is low, the GAR training will introduce a higher modification rate, as is the case for DeepSeek-Prover. But if the modification rate is high in the base model, the soft penalty will control it in a reasonable range. These findings prove the improvement of the GAR-trained model from another side. We will add this data to the next version of our paper to clarify that this stable statement modification represents robust reasoning rather than instruction following failure.
>
> We hope that this rebuttal addresses your concerns, and we are grateful for your recognition of the novelty of our work. Your feedback encourages us to continue refining our research.
>
> Best Regards,
>
> Submission #7861 Author Team
>
> **References:**
>
> [1] Dong, K., & Ma, T. (2025). Stp: Self-play llm theorem provers with iterative conjecturing and proving. *arXiv preprint arXiv:2502.00212*.
>
> [2] Lin, Y., Tang, S., Lyu, B., Wu, J., Lin, H., Yang, K., ... & Jin, C. (2025). Goedel-prover: A frontier model for open-source automated theorem proving. *arXiv preprint arXiv:2502.07640*.
>
> [3] Wei, Y., Wang, Z., Liu, J., Ding, Y., & Zhang, L. (2023). Magicoder: Empowering code generation with oss-instruct. *arXiv preprint arXiv:2312.02120*.
>
> [4] Guo, D., Yang, D., Zhang, H., Song, J., Zhang, R., Xu, R., ... & He, Y. (2025). Deepseek-r1: Incentivizing reasoning capability in llms via reinforcement learning. *arXiv preprint arXiv:2501.12948*.

---

> > ### Author Response · Authors · 2025-11-26
> > **Follow-up to reviewer BPXJ**
> >
> > Dear Reviewer BPXJ,
> >
> > Thank you for your constructive feedback and insightful suggestions. Your positive remarks about our work are very encouraging, particularly your recognition of our novel formulation, promising empirical gains, clear and reproducible methods, and the potential for general application of our proposed methods.
> >
> > We are grateful for your efforts in identifying potential flaws in our work, which we have attempted to address in our response above by: **(1) providing a further ablation study on frozen fuser setup; (2) providing the SFT only baseline; (3) comparing with more data generation methods; (4) clarifying the statement fuser study; (5) clarifying the design principle of statement modification penalty; (6) clarifying the efficiency study; (7) providing further analysis on statement modification rates and prompt shift**.
> >
> > With the discussion period processing, we would greatly appreciate any further comments you may have regarding our response, as we are eager to address any additional issues. If you find that our response has adequately addressed your concerns, we would really appreciate it if you could consider raising the score.
> >
> > We fully understand that this may be a busy period for you, and we sincerely appreciate your efforts in helping us improve our work. We look forward to receiving any additional feedback you may have.
> >
> > Best Regards,
> >
> > Submission #7861 Author Team

---

### Official Review · Reviewer_TB6c · 2025-10-26

**Soundness:** 3
**Presentation:** 3
**Contribution:** 2
**Rating:** 6
**Confidence:** 3

**Summary:**

This work introduces a novel RL framework (GAR) that trains not only a solver, but also a fuser designed to generate new problems suitable for the solver's current training stage. To be specific, in the context of automated theorem proving, the prover (solver) is trained to solve problems. Concurrently, the fuser attempts to generate problems that are provable in Lean (by the current provers) but possess a relatively low pass rate, thus making them "difficult but approachable." The fuser is also trained during the RL process, creating an adversarial environment. The experimental results demonstrate the effectiveness of this method.

**Strengths:**

The overall quality and clarity of this work are good. Its motivation and methodology are described in detail. The proposed method also demonstrates significance, as it is potentially applicable to domains beyond automated theorem proving.

**Weaknesses:**

Regarding novelty, the idea of generating problems with appropriate difficulty levels has been explored by prior works, such as STP and Goedel-Prover-V2. STP, in particular, also trained a model to generate new problems in an adversarial way. The main difference appears to be that this work uses RL, whereas STP used expert iteration (SFT). This overlap weakens the originality and significance of the current paper.

Additionally, some experimental results in Table 1 are questionable. It appears that the reported results for prior works are all much lower than those originally published in their respective papers.

**Questions:**

Regarding Table 1, could you please explain why your reported results for previous works (Kimina, Deepseek, Goedel) are all significantly lower than those in their original papers? Did you encounter any issues with your experimental setup, such as a different Lean version, that might account for this mismatch?

---

> ### Author Response · Authors · 2025-11-21
> **Response to Review from TB6c**
>
> Dear Reviewer TB6c,
>
> We offer our sincere gratitude for your thoughtful review and constructive comments. We deeply appreciate your recognition of the significance and potential of our work. Below, we provide detailed responses to the concerns you raised.
>
> **W1. Novelty compared to STP and Goedel-Prover-V2:**
>
> We appreciate the opportunity to clarify the distinctions between our work and prior approaches through your raised concern. While GAR, STP, and Goedel-Prover-V2 all optimize the training dataset as the training proceeds, our methodology differes from previous works in *when and how* such optimization occurs. Specifically, GAR dynamically updates the problem set during the online RL phase, whereas the other methods rely on updates during the SFT phase, making GAR’s optimization more efficient.
>
> 1. **Compared to Goedel-Prover-V2:** The statement generation in Goedel-V2 relies on a frozen large model to synthesize data for SFT. Crucially, during the RL phase, their statement set remains static. This lack of dynamic updates of statements based on the prover’s evolving capabilities may lead to suboptimal performance and efficiency as the prover outpaces the fixed problem set. In contrast, GAR continuously updates the statement fuser via adversarial RL, ensuring the generated statements remain progressively challenging and aligned with the proer’s current skill level during the RL phase, leading to better empirical performance.
> 2. **Compared to STP:** While STP trains a conjecture model based on the prover’s feedback, it relies on the expert iteration framework based on offline SFT. This process is inherently inefficient because it requires generating a massive volume of training data in a single iteration to achieve effective SFT training. According to [1], they generate **75,000 conjectures per iteration**, which is larger than the entire statement set for our generation. Conversely, GAR operates within an **online RL cycle**. This allows the fuser and prover to continuously update more efficiently, achieving superior performance without the computational burden of the large-scale per-iteration data required by STP.
>
>     We will include this explicit comparison in the final version of the paper to enhance clarity.
>
> **W2. & Q1. Discrepancies in experimental results:**
>
> We thank you for raising this point and would like to clarify it. The difference in performance between Goedel-Prover-V2 and DeepSeek-Prover in our paper and in their paper is due to the **maximum context length constraint.**
>
> As noted in Line 377 of the initial version of our paper, we limit the context length for all experiments to **16,384 tokens** due to computational constraints. In contrast, the official HF repo for Goedel-Prover-V2-8B [2] indicates a context length of 40,960, and their official GitHub [3] repo uses 131,072 context length. The DeepSeek-Prover-V2-8B’s official HF repo applies a max sequence length of 65,536 tokens. Our initial test indicates that extending the context to 40,960 tokens for the MiniF2F-Test pass@32 evaluation on Goedel-Prover-V2-8B requires **200 A100-80G hours,** which makes the entire experiment cost more than  2,000 hours. Our computing resource is unable to support such a cost. The reduced context length leads to truncation of long reasoning CoT and resulting in lower scores compared to the original papers.
>
> Regarding Kimina-Prover-Preview-7B, we wish to clarify that our reported result in Table 1 is identical to the results reported in their original paper [5]. We utilized the same context length and evaluation settings as [5] indicates and reached similar results. Thus, we directly use the results in their paper. We will add a detailed note regarding these configuration differences in the next version of our paper to prevent confusion.
>
> We hope that this rebuttal addresses your concerns, and we are grateful for your recognition of the contribution of our work. Your feedback encourages us to continue refining and extending this line of research.
>
> Best Regards,
>
> Submission #7861 Author Team

---

> > ### Author Response · Authors · 2025-11-21
> > **References for our response**
> >
> > **References:**
> >
> > [1] Dong, K., & Ma, T. (2025). Stp: Self-play llm theorem provers with iterative conjecturing and proving. *arXiv preprint arXiv:2502.00212*.
> >
> > [2] Lin, Y., Tang, S., Lyu, B., Yang, Z., Chung, J. H., Zhao, H., ... & Jin, C. (2025). *Goedel-Prover-V2-8B* [Computer software]. Hugging Face. https://huggingface.co/Goedel-LM/Goedel-Prover-V2-8B
> >
> > [3] Lin, Y., Tang, S., Lyu, B., Yang, Z., Chung, J. H., Zhao, H., ... & Jin, C. (2025). *Goedel-Prover-V2* (Version 2.0) [Computer software]. GitHub. https://github.com/Goedel-LM/Goedel-Prover-V2
> >
> > [4] Ren, Z. Z., Shao, Z., Song, J., Xin, H., Wang, H., Zhao, W., ... & Ruan, C. (2025). *DeepSeek-Prover-V2-7B* [Computer software]. Hugging Face. https://huggingface.co/deepseek-ai/DeepSeek-Prover-V2-7B
> >
> > [5] Wang, H., Unsal, M., Lin, X., Baksys, M., Liu, J., Santos, M. D., ... & Li, J. (2025). Kimina-prover preview: Towards large formal reasoning models with reinforcement learning. *arXiv preprint arXiv:2504.11354*.

---

> > > ### Author Response · Authors · 2025-11-26
> > > **Follow-up to reviewer TB6c**
> > >
> > > Dear Reviewer TB6c,
> > >
> > > Thank you for your constructive feedback and insightful suggestions. Your positive remarks about our work are very encouraging, particularly your recognition of our:
> > >
> > > 1. Good quality and clarity
> > > 2. Detailed description of motivation and methodology
> > > 3. The potential for further application of our work.
> > >
> > > We are grateful for your efforts in identifying potential flaws in our work, which we have attempted to address in our response above by **performing explicit comparisons with related works** and **clarifying our experiment design**. With the discussion period processing, we would greatly appreciate any further comments you may have regarding our response, as we are eager to address any additional issues.
> > >
> > > If you find that our response has adequately addressed your concerns, we would really appreciate it if you could consider raising the score.
> > >
> > > We fully understand that this may be a busy period for you, and we sincerely appreciate your efforts in helping us improve our work. We look forward to receiving any additional feedback you may have.
> > >
> > > Best Regards,
> > >
> > > Submission #7861 Author Team

---

### Official Review · Reviewer_oK7C · 2025-11-01

**Soundness:** 4
**Presentation:** 4
**Contribution:** 3
**Rating:** 8
**Confidence:** 5

**Summary:**

The paper proposes a variant of self-play theorem proving (conjecturer-prover adversarial training) where the conjecturer fuses two statements to create a statement with higher difficulty "Composability" is a frequently observed problem in LLMs, and the method directly tackles this issue. With small budgets of additional training, the method improves the initial already highly optimized models by a significant margin.

**Strengths:**

The method is sensible, the numbers are very good, the paper is very clearly written.
The scope is limited because there are many ways of generating synthetic statements and there was no comparison to other synthetic data methods (STP, Magicoder), but it appears that the authors have found a particularly effective method for this.

**Weaknesses:**

No comparisons to other synthetic data generation methods.

**Questions:**

- Can you expand on the issue of <analysis> ?
- Why the statement modification loss and not simply freezing the statement (and punishing any modifications as fully incorrect)? Doesn't this give an optimal policy that is not intended? Why doesn't that happen in practice?

Nits:
eq 7: r_{ijk} needs a correctness term instead of 1?
Alg. 1: typo in line 7

---

> ### Author Response · Authors · 2025-11-13
> **Potentional formatting issue of question 1**
>
> Dear Reviewer oK7C,
>
> We sincerely appreciate your recognition of our work and your valuable feedback.
> However, while preparing our rebuttal, we noticed that question 1 contains `[object Object]`, which appears to be a formatting issue, and we do not understand. Could you clarify what specific issue or aspect you would like us to address?
>
> We are eager to provide a thorough response once we better understand your question.
>
>
> Best,
> 7861 Author Team

---

> > ### Comment · Reviewer_oK7C · 2025-11-13
> >
> > sorry, issue with the tag formatting, edited

---

> ### Author Response · Authors · 2025-11-21
> **Response to Review from oK7C**
>
> Dear Reviewer oK7C,
>
> We thank you so much for your positive assessment and recognition of the soundness and empirical effectiveness of the GAR framework. We appreciate your constructive feedback, which has helped us to enhance the soundness of our paper. We respond to your concerns as follows:
>
> **W1. Compared to other synthetic data generation methods:**
>
> We agree that comparing different synthesis strategies is essential for the completeness of our work. Following your suggestion, we replaced our Statement Fusion module with a single-problem evolution rather than fusing pairs, following the Magicoder pattern. We trained the Goedel-Prover-V2-8B model for three iterations using the same setup as in the original paper. The pass@32 performance of MiniF2F-Test is as follows:
>
> | Model Type | MiniF2F-Test pass@32 |
> | --- | --- |
> | Goedel-Prover-V2-8B | 77.87% |
> | MagicCoder-GAR Goedel-V2 | 75.41% |
> | GAR Goedel-V2 | **80.33%** |
>
> The result indicates that replacing the fusion process with the Magicoder-style problem evolving strategy results in a 2.46% performance drop compared to the base model. We conclude this degradation to the limited capability of Qwen3-8B to create more advanced problems with only one reference. It may not provide sufficient difficulty gradient for the prover, leading to training failure. In contrast, our fusion approach creates a more effective implicit curriculum by combining distinct concepts from multiple problems, thereby validating the design of the Statement Fuser. We will add this experiment to the next version of our paper to make the ablation study more complete.
>
> **Q1. Details regarding the `<analysis>`  block:**
>
> We appreciate that you raised this point. To address this problem, we have added a detailed case study comparing the standard Long CoT in Qwen3-8B with our reinitialized `<analysis>` CoT. A detailed case study can be found in [here](https://anonymous.4open.science/r/GAR_reinitLongCoT_example-4748/README.md).
>
> In summary, applying standard Long CoT to the open-ended statement fusion problem will lead to the following issues:
>
> 1. Problem misidentification: The model frequently misidentifies the mathematical domain (e.g., confusing set theory with unit conversion)
> 2. Inefficient exploration: The reasoning trace in the original Long CoT often aimlessly wanders and suffers from excessive repetition, failing to converge on a valid problem structure.
> 3. Relevance failure: The resulting statements often detach completely from the source material.
>
> On the other hand, the reinitialized Long CoT using `<analysis>` block acts as a soft restart of the thinking process, forcing the model to discard irrelevant context and explicitly structure its fusion strategy. This mechanism ensures the generated theorems remain solvable and grounded in the source concepts.
>
> **Q2 Why not completely ban the statement modification behavior:**
>
> We thank the reviewer for raising the question. Please kindly refer to line 243 of the original draft of our paper, where we indicate that modern provers utilize Long CoT for self-correction, which is connected to the capability of fixing the problem that the model considers to be unsolvable. Imposing a strict ban treats these valid reasoning steps as failures, thereby suppressing the model’s capability ot self-correct.
>
> To confirm this, we conducted an ablation study where we strictly prohibited statement modifications during the training of Goedel-Prover-V2. The resulting model only achieved a 77.46% pass@32 rate for the MiniF2F-Test, underperforming the base model. It indicates our soft penalty strikes a necessary balance: it is high enough to discourage severe reward hacking but low enough to preserve the model’s self-correction capability.
>
> Additionally, we thank you for catching the typo in Eq. (7). We will correct this and any other typo we identified in the next version of our paper.
>
> We hope these responses adequately address your concerns. We are grateful for your endorsement of our contribution. Your feedback encourages us to further improve and extend research in the field.
>
> Best Regards,
>
> Submission #7861 Author Team

---

> > ### Author Response · Authors · 2025-11-26
> > **Follow-up to reviewer oK7C**
> >
> > Dear Reviewer oK7C,
> >
> > Thank you for your constructive feedback and insightful suggestions. Your positive remarks about our work are very encouraging, particularly your recognition of our sensible methodology, good empirical performance, clear paper writing, and the effective method we found.
> >
> > We are grateful for your efforts in identifying potential flaws in our work, which we have attempted to address in our response above by **providing analysis with other data generation methods, further elaborating the functions of `<analysis>` block, and clarification of the design principle for statement modification penalty**. With the discussion period processing, we would greatly appreciate any further comments you may have regarding our response, as we are eager to address any additional issues.
> >
> > We would greatly appreciate it if you would consider that our rebuttal is sufficient for you to increase the score. We fully understand that this may be a busy period for you, and we sincerely appreciate your efforts in helping us improve our work. We look forward to receiving any additional feedback you may have.
> >
> > Best Regards,
> >
> > Submission #7861 Author Team

---

### Official Review · Reviewer_Kc9L · 2025-11-01

**Soundness:** 3
**Presentation:** 3
**Contribution:** 2
**Rating:** 6
**Confidence:** 4

**Summary:**

The paper introduces GAR (Generative Adversarial Reinforcement Learning) for formal theorem proving in Lean. Unlike traditional expert iteration or RL for theorem proving with a fixed statement set, GAR dynamically generate new statements at the right difficulty level for the prover, forming an implicit curriculum. The method consists of a composer that generates new statements by fusing existing statements and a prover. Both components are trained with RL. The composer is incentivized to generate statements that are challenging but provable for the current prover. Experiments on MiniF2F and ProofNet shows improvements to the prover's capability.

**Strengths:**

* The paper addresses a critical bottleneck in current neural theorem provers, i.e., the reliance on a fixed and limited set of theorem statement. Having the model conjecturing and proving new statement is a promising direction. The design space here is quite big, and GAR represents one reasonable method in this space.
* Experiments show good improvements on MiniF2F and ProofNet.

**Weaknesses:**

* GAR is not the first attempt to augment the set of formal statements. For example, the scaffolded data synthesis in Goedel-Prover-V2 tries to generate simpler variants of hard problems and harder variants of simple problems. STP also generates new statements based on existing statements. The fuser in this paper is different from those methods, but it would be great if the authors can make the comparison more explicit.
* MiniF2F is already saturated. SOTA methods are close to 100%. Experiments on PutnamBench would make this paper much stronger.

**Questions:**

N/A

---

> ### Author Response · Authors · 2025-11-21
> **Response to Review from Kc9L**
>
> Dear Reviewer Kc9L,
>
> Thank you for your thoughtful review and constructive suggestions. We appreciate your recognition of our work’s contribution to addressing the fixed-statement bottleneck in the LLM theorem proving. We provide the detailed responses to your concerns below:
>
> **W1 Explicit comparison with Goedel-Prover-V2 and STP:**
>
> We appreciate your suggestion to demonstrate the distinct advantages of GAR’s adversarial RL framework compared to prior data synthesis methods. The explicit comparison is as follows:
>
> 1. **Compared to Goedel-Prover-V2:** The statement generation in Goedel-V2 relies on a frozen large model to synthesize data for SFT. Crucially, during the RL phase, their statement set remains static. This lack of dynamic updates of statements based on the prover’s evolving capabilities may lead to suboptimal performance and efficiency as the prover outpaces the fixed problem set. In contrast, GAR continuously updates the statement fuser via adversarial RL, ensuring the generated statements remain progressively challenging and aligned with the proer’s current skill level during the RL phase, leading to better empirical performance.
> 2. **Compared to STP:** While STP trains a conjecture model based on the prover’s feedback, it relies on the expert iteration framework based on offline SFT. This process is inherently inefficient because it requires generating a massive volume of training data in a single iteration to achieve effective SFT training. According to [1], they generate **75,000 conjectures per iteration**, which is larger than the entire statement set for our generation. Conversely, GAR operates within an **online RL cycle**. This allows the fuser and prover to continuously update more efficiently, achieving superior performance without the computational burden of the large-scale per-iteration data required by STP.
>
> **W2 Additional experiment results on PutnamBench:**
>
> We agree that the MiniF2F is becoming saturated, which is why we also test the performance of our model on more challenging ProofNet in the paper and achieve a **3.23% absolute** improvement. We also agree that testing the model on PutnamBench is necessary. Thus, we evaluated the GAR-trained DeepSeek-Prover-V2-7B on PutnamBench using the pass@32 metric. The results are as follows:
>
> | Model Type | PutnamBench pass@32 |
> | --- | --- |
> | DeepSeek-Prover-V2-7B | 22/660 |
> | GAR DeepSeek-Prover | 26/160 |
>
> We can see that the GAR-trained model solves four additional problems compared to the base model on this highly difficult benchmark. This consistent improvement across MiniF2F, PutnamBench, and ProofNet demonstrates the robustness of the implicit curriculum established by GAR.
>
> We hope these responses address your concerns. We are sincerely grateful for your feedback, which has greatly helped improve the clarity and completeness of our work. The additional experiment results, as well as explicit comparison, will be added to the next version of our paper.
>
> Best Regards,
>
> Submission #7861 Author Team
>
> **References:**
>
> [1] Dong, K., & Ma, T. (2025). Stp: Self-play llm theorem provers with iterative conjecturing and proving. *arXiv preprint arXiv:2502.00212*.

---

> > ### Author Response · Authors · 2025-11-26
> > **Follow-up to reviewer Kc9L**
> >
> > Dear Reviewer Kc9L,
> >
> > Thank you for your constructive feedback and insightful suggestions. Your positive remarks about our work are very encouraging, particularly your recognition of our:
> >
> > 1. Our generative adversarial solution for addressing the critical problems of a fixed and limited set of theorem statements in the field.
> > 2. The good performance in the MiniF2F and ProofNet datasets for GAR-trained models.
> >
> > We are grateful for your efforts in identifying potential flaws in our work, which we have attempted to address in our response above by **performing explicit comparisons with related works** and **providing additional experiments on PutnamBench**. With the discussion period processing, we would greatly appreciate any further comments you may have regarding our response, as we are eager to address any additional issues.
> >
> > If you find that our response has adequately addressed your concerns, we would really appreciate it if you could consider raising the score.
> >
> > We fully understand that this may be a busy period for you, and we sincerely appreciate your efforts in helping us improve our work. We look forward to receiving any additional feedback you may have.
> >
> > Best Regards,
> >
> > Submission #7861 Author Team

---

### Author Response · Authors · 2025-12-02
**Discussion Summary for AC and SAC**

Dear AC and SAC

We sincerely appreciate your tremendous efforts in handling this urgent OpenReview information leak incident. To speed up your process, we would like to provide some critical information about our submission.

**Paper Overview:**

Our paper introduces GAR (Generative Adversarial Reinforcement Learning), a framework that jointly trains the statement fuser and the theorem prover through adversarial co-evolution. Unlike prior methods that rely on static problem sets during RL training, GAR dynamically generates statements at appropriate difficulty levels, creating an implicit curriculum that continuously provides challenging problems to the prover.

**Discussion period summary:**

All the reviewers think positively of our work. For example

1. Reviewer Kc9L thinks our work addresses a critical bottleneck in theorem proving, and experiments show good improvements.
2. Reviewer oK7C considers our methods sensible; the numbers are very good, and the paper is very clearly written. oK7C also recognizes the effectiveness of our methodology
3. Reviewer TB6c considers our work to be of good quality and clarity, recognizes the significance of our proposed methods, and the potential to generalize.
4. Reviewer BPXJ praised the novelty of our work, the measurable and consistent empirical gains. BPXJ also considers our methods reproducible and potentially generalizable to other fields.

During the rebuttal period, we also intended to clarify certain points and conduct additional experiments to make our work more comprehensive. We conducted new experiments, including PutnamBench evaluation, frozen-fuser ablation, MagicCoder-style problem synthesis comparison, and SFT-only base model validation.

Furthermore, we clarified our work by making an explicit comparison between GAR and STP/Goedel-V2 (online RL vs. offline SFT efficiency), explaining the context length differences that account for the discrepancies in Table 1, and performing an analysis of statement modification rates.

During the discussion period, all ratings remained, and no reviewer raised any additional questions. Because of this, we have uploaded a revised version of our paper, which includes all the clarifications and additional experiments promised during the discussion period in the Appendix section.

We sincerely appreciate your efforts in reviewing our paper, as well as the comments from the reviewers and our responses.

Best Regards

Submission #7861 Author Team

---

### Meta-Review · Area_Chair_4L3D · 2026-01-07

**Summary:**

This paper proposes an adversarial RL framework for theorem proving, in which a prover interacts with a proposer of theorems, which is trained to propose provable but challenging statements. The main points made by the reviewers are the following.

Strengths:
1. Well-motivated/interesting idea (Kc9L, oK7C, 7B6c, BPXJ)
2. Solid experimental results (Kc9L, oKtC, BPXJ)
3. Good writing (oK7C, 7B6c, BPXJ)

Weaknesses/concerns:
1. Novelty over / comparison with other attempts to grow the problem training set (Kc9L, oK7C, TB6c)
2. Requests for harder dataset (Kc9L)
3. Question on "analysis" chain of thought and statement modification loss (oK7C, BPXJ)
4. Question about results not matching prior work (TB6c)
5. Request for studies on training only the prover, SFT baseline, fuser design, difficulty analysis (BPXJ)
6. Efficiency concerns (BPXJ)

The authors responded to all of these concerns, as detailed below, and think that the reviewers -- already all recommending acceptance -- would have been satisfied with the responses. I therefore recommend acceptance and suggest that the authors incorporate all review feedback and rebuttal content into the final version.

**Reviewer Concerns:**

1. Answered in rebuttal by new experiment and discussion.
2. Addressed in rebuttal. (NB. Is the improvement from 22 to 26 problems solved really "four additional problems", i.e., the 22 are a subset of the 26? This affects the significance.)
3. Addressed in rebuttal, along with an interesting new analysis in the response to BPXJ.
4. The rebuttal clarified that the difference is due to differences in the context length constraint. This makes the result less satisfying (as ideally we would compare to the best models as they were run in prior work), but it is perhaps more unsatisfying that the leading work requires such high context lengths (and therefore budgets), and the improvements from the proposed method are meaningful and plainly visible even at lower context lengths.
5. All answered in the rebuttal with new experiments for each request.
6. The authors acknowledge the efficiency concern with RL-trained base models, but argue that SFT-only models are unreasonably weak base models. However, an efficiency/cost comparison against the other methods for growing the training set, all using the same RL-trained base model, would be useful to include in the revision.

**Reviewer Scores:**

The original scores were 6 (Kc9L), 8 (oK7C), 6 (TB6c), 6 (BPXJ). None of the reviewers responded to the rebuttals. Given the responses, it is likely that all three of the lower scores would have been increased to 8.

---

### Decision · Program_Chairs · 2026-01-26

Accept (Poster)